



# Concentrations and biosphere-atmosphere fluxes of inorganic trace gases and associated ionic aerosol counterparts over the Amazon rainforest

Robbie Ramsay[1,2], Chiara F Di Marco[1], Matthias Sörgel[3,*], Mathew R Heal[2], Samara Carbone[4], Paulo Artaxo[5], Alessandro C de Araùjo[6], Marta Sá[7], Christopher Pöhlker[8], Jost Lavric[9], Meinrat O. Andreae[3,10], and Eiko Nemitz[1]

[1]UK Centre for Ecology and Hydrology (UKCEH), Bush Estate, Penicuik, EH26 0QB, UK
[2]School of Chemistry, University of Edinburgh, Joseph Black Building, David Brewster Road, Edinburgh EH9 3FJ, UK
[3]Biogeochemistry Department, Max Planck Institute for Chemistry, 55128 Mainz, Germany
[4]Federal University of Uberlândia, Agrarian Sciences Institute, Uberlândia, MG, Brazil
[5]Instituto de Física, Universidade de São Paulo, São Paulo, Brazil
[6]Empresa Brasileira de Pesquisa Agropecuária (EMBRAPA), Belèm-PA, CEP 66095-100, Brazil
[7]Large Scale Biosphere-Atmosphere Experiment in Amazonia (LBA), Instituto Nacional de Pesquisas da Amazonia (INPA), Manaus-AM, CEP 69067-375, Brazil
[8]Atmospheric Chemistry Department, Max Planck Institute for Chemistry, Mainz, Germany
[9]Department Biogeochemical Systems, Max Planck Institute for Biogeochemistry, Jena, Germany
[10]Scripps Institution of Oceanography, University of California San Diego, La Jolla, CA, USA
*now at: Atmospheric Chemistry Department, Max Planck Institute for Chemistry, Mainz, Germany

**Correspondence:** Eiko Nemitz (en@ceh.ac.uk)

**Abstract.** The Amazon rainforest presents a unique, natural laboratory for the study of surface-atmosphere interactions. Its alternation between a near-pristine, marine-influenced atmosphere during the wet season, and a vulnerable system affected by periodic intrusions of anthropogenic pollution during the dry season, provides an opportunity to investigate some fundamental aspects of boundary-layer chemical processes. This study presents the first simultaneous hourly measurements of concentra-
tions, fluxes and deposition velocities of the inorganic trace gases $NH_3$, $HCl$, $HONO$, $HNO_3$ and $SO_2$ and their water-soluble aerosol counterparts $NH_4^+$, $Cl^-$, $NO_2^-$, $NO_3^-$ and $SO_4^{2-}$ over the Amazon. Species concentrations were measured in the dry season (from 6 October to 5 November 2017), at the Amazon Tall Tower Observatory (ATTO) in Brazil, using a two-point gradient, wet-chemistry instrument (Gradient of Aerosols and Gases Online Registration, GRAEGOR) sampling at 42 m and 60 m. Fluxes and deposition velocities were derived from the concentration gradients using a modified form of the aerodynamic
gradient method corrected for measurement within the roughness sub-layer. Findings from this campaign include observations of elevated concentrations of $NH_3$ and $SO_2$ partially driven by long-range transport (LRT) episodes of pollution, and the substantial influence of coarse $Cl^-$ and $NO_3^-$ particulate on overall aerosol mass burdens. From the flux measurements, the dry season budget of total reactive nitrogen dry deposition at the ATTO site was estimated as -2.9 kg N $ha^{-1}a^{-1}$. $HNO_3$ and $HCl$ were deposited continuously at a rate close to the aerodynamic limit. $SO_2$ was deposited with an average daytime surface
resistance ($R_c$) of 28 s $m^{-1}$, whilst aerosol components showed average surface deposition velocities of 2.8 and 2.7 mm $s^{-1}$ for $SO_4^{2-}$ and $NH_4^+$. Deposition rates of $NO_3^-$ and $Cl^-$ were higher at 7.1 and 7.8 mm $s^{-1}$, reflecting their larger average size.



The exchange of NH$_3$ and HONO was bi-directional, with NH$_3$ showing emission episodes in the afternoon and HONO in the early morning hours. This work provides a unique dataset to test and improve dry deposition schemes for these compounds for tropical rain forest, which have typically been developed by interpolation from conditions in temperate environments. A

future campaign should focus on making similar measurements in the wet season in order to provide a complete view of the annual pattern of inorganic trace gas and coarse aerosol biosphere-atmosphere exchange over tropical rainforest.

## 1 Introduction

The Amazon rainforest is one of the last remaining wildernesses on Earth, which—-through a select combination of environmental and geographical factors—acts as a critical, living driver of global climate (Malhi et al., 2008). It is a vast region of

near-undisturbed verdant growth, covering almost 60% of the total land area of Brazil, and constituting almost 40% of global tropical forest cover (Baccini et al., 2012). It stores an estimated 160 Pg of organic carbon in its soils (Gloor et al., 2012), and harbours an immense atmospheric oxidative capacity driven by a powerful hydrological cycle (Lelieveld et al., 2008). The strong coupling between the forest and the atmosphere, and the sensitive feedbacks between them that regulate atmospheric composition, has earned the Amazon rainforest the sobriquets of the "Green Ocean" (Martin et al., 2016; Roberts et al.,

2001; Williams et al., 2002) and the "biogeochemical reactor" (Pöhlker et al., 2012; Andreae, 2001). It is therefore not only a near pristine microcosm of the pre-Anthropocene, but also acts as a continental "natural laboratory" to study unmodified surface-atmosphere exchange processes.

However, the combination of global climate change and the intensification of human development within and on the periphery of the rainforest has left the Amazonian biome in a precarious situation (Davidson et al., 2012). Emissions of pollutants from

agricultural activities, biomass burning and deforestation in the vicinity of the rainforest can perturb its surface-atmosphere exchange processes (Ganzeveld and Lelieveld, 2004) and cause changes in the local, regional, and even global climate (Lenton et al., 2008).

While measurements of the atmospheric composition and surface-atmosphere exchanges process of the Amazon rainforest have been conducted since the late 1980s (e.g. Andreae and Andreae, 1988; Artaxo et al., 1993; Martin et al., 2010a) , there

remain significant knowledge gaps. Fundamental questions such as the magnitude of inorganic trace gas fluxes and the chemical speciation of coarse aerosols remain partially unanswered. A pressing need is for more baseline measurements of gases and aerosols in order to quantify the impact of anthropogenic changes.

This latter point has been addressed by the establishment of the Amazon Tall Tower Observatory (ATTO). Located in a pristine rainforest site 150 km NE of the city of Manaus, the site provides the baseline measurements of meteorology, trace

gases and aerosol required to quantify the impact of natural and anthropogenic change (Andreae et al., 2015). Recent output has included a long-term overview of cloud condensation nuclei over the Amazon Rainforest (Pöhlker et al., 2016, 2018), observations of the enhancement of deep convection over the rainforest by ultrafine particles (Fan et al., 2018), and the influence of African volcanic emissions on long-range transport of pollutants to the ATTO site (Saturno et al., 2018b; Holanda et al., 2020). However, several inorganic trace gases and their aerosol counterparts are currently not routinely measured due to the





intense labour and resource requirements. The aim of this work was to make such measurements via an intensive observation campaign; in particular, to derive the first time-series of simultaneous flux measurements of these species at this tropical rainforest site.

The gas species of interest include ammonia ($NH_3$), nitrous acid (HONO), hydrogen chloride (HCl), nitric acid ($HNO_3$) and the precursor to atmospheric sulfuric acid, sulfur dioxide ($SO_2$). As the primary basic gas in the atmosphere, $NH_3$ is

important as the precursor of various ammonium salts, particularly $NH_4NO_3$, formed by the temperature and humidity dependent reaction between $NH_3$ and $HNO_3$. These salts act as light-scattering aerosols in the atmosphere, altering the Earth's total albedo and consequently affecting regional and global climate (Fiore et al., 2015). Depending on environmental conditions, ammonium salts can be particularly long lived, and their eventual decomposition above nitrogen-limited ecosystems—-such as the Amazon rainforest-—can lead to disturbances in soil fertility, vegetation composition, and pollution of groundwater

sources (Fowler et al., 2013). The dynamic equilibrium between $NH_3$, $HNO_3$, and $NH_4NO_3$ makes it difficult to determine the surface-atmosphere exchange of the individual members of the triad. To date, very few simultaneous measurements of each component in real time and with high time resolution exist (Ramsay et al., 2018; Trebs et al., 2006; Twigg et al., 2011; Wolff et al., 2010b), and none over tropical rainforest.

Measurements of HONO are also critically required due to its potential contribution to atmospheric hydroxyl radical (OH)

concentrations. The OH radical is the primary daytime oxidant in the Amazon rainforest, and is principally formed via ultra-violet (UV) photodissociation of ozone in the presence of water vapour. In the tropics, where there is intense solar radiation and high humidity, concentrations of the OH radical are elevated relative to the global median (Kuhn et al., 2007; Lelieveld et al., 2002; Taraborrelli et al., 2012). The photodissociation of HONO also yields OH, and so may make a crucial contribution to sustaining the overall oxidative capacity above the Amazon rainforest. Non-negligible concentrations of HONO have been

reported at urban (Lee et al., 2016), agricultural (Laufs et al., 2017; Twigg et al., 2011) and rural European forest sites (Sörgel et al., 2011) but there are currently no published measurements of HONO concentrations or fluxes above tropical rainforest.

There is also a need for better quantification of aerosols, particularly chemically-speciated particulate matter, aerosol deposition velocities, and surface-atmosphere exchange behaviour. The majority of aerosol measurements at the ATTO site have so far focused on the sub-micron ($< PM_1$) size fraction, reflecting the importance of these particles in seeding cloud condensa-

tion nuclei and their seasonal and temporal variability driven by biomass burning (Artaxo et al., 2013; Martin et al., 2010b; Pöschl et al., 2010; Pöhlker et al., 2016, 2018). Studies of coarse particles are limited (Talbot et al., 1990; Moran-Zuloaga et al., 2018; Whitehead et al., 2016), but have confirmed that coarse fraction aerosols are driven by the transport of dust, sea salt, primary biogenic aerosols, and particles transported in smoke from biomass burning. While number concentrations and chemically-speciated sub-micron aerosol particles have been measured, there are currently no flux or deposition velocity data

for chemically-speciated fine or coarse-mode particles for the Amazon rainforest.

Determination of concentrations and fluxes of trace gases and aerosol components requires precise, high-time resolution measurements. Instruments must also be sensitive to the often very low concentrations in remote locations such as the Amazon rainforest. Compounding these requirements is the potential impact of gas-particle interactions that must be considered for accurate descriptions of surface-atmosphere exchange. This requires concurrent, multi-species measurements.





Development in automated wet-chemistry instruments has led to the construction of the Gradient of Aerosols and Gases Online Registration (GRAEGOR), which is capable of simultaneously measuring the concentrations of the inorganic trace gases $NH_3$, HCl, HONO, $HNO_3$ and $SO_2$ and their associated water-soluble aerosol counterparts $NH_4^+$, $Cl^-$, $NO_2^-$, $NO_3^-$ and $SO_4^{2-}$ at two separate heights at hourly resolution (Thomas et al., 2009). Fluxes for each of these species can then be derived from the two concentrations using a modified version of the aerodynamic gradient method (AGM), from which hourly

values for the deposition velocities ($V_d$) of each species can also be determined. A number of campaigns have now confirmed the suitability of the GRAEGOR for measuring vertical concentration gradients and fluxes of these trace gases and aerosol components (Ramsay et al., 2018; Thomas et al., 2009; Twigg et al., 2011; Wolff et al., 2010b).

The overall aim of this study was to resolve some of the knowledge gaps in the biosphere-atmosphere exchange of inorganic trace gases and aerosols to and from tropical rainforest. We present here the concentrations, fluxes and deposition velocities

of the trace gases $NH_3$, HCl, HONO, $HNO_3$ and $SO_2$ and their associated aerosol counterparts $NH_4^+$, $Cl^-$, $NO_2^-$, $NO_3^-$ and $SO_4^{2-}$ as measured by the GRAEGOR wet-chemistry, two-point gradient system during a period of the 2017 dry season at the ATTO site. Using supplementary measurements of non-refractory, chemically differentiated sub-micron aerosol and concentrations of atmospheric equivalent black carbon, we elucidate the lifetime, behaviour and origins of the measured trace gases and aerosols.

## 2 Methodology

### 2.1 Site Description

The measurements presented here are from an intensive observation campaign conducted at the ATTO site from 6 October to 5 November 2017. Situated on a level plateau located 12 km northwest of the Uatumã River, the ATTO site lies 150 km northeast of the Manaus urban region. The site is located within the Amazon Time Zone (UTC – 4 hours). All times presented

in this work are given as local time. The vegetation is composed of dense, undisturbed upland rainforest (*terra firme*), with a rich tree diversity ($\approx$140 tree species $ha^{-1}$) (Andreae et al., 2015). Based on the height of the tallest trees, the canopy height ($h_c$) is 37.5 m (Chor et al., 2017). The site lies within the central Amazonian region, and experiences an annual oscillation between wet and dry seasons with transitional periods, driven by the position of the intertropical convergence zone (ICTZ). The wet season, typically lasting between February and May when the ICTZ is south of the ATTO site, is characterised by

north-easterly (NE) trade winds bringing air masses from the North Atlantic. These travel over hundreds of kilometres of untouched rainforest, leading to near pristine atmospheric conditions. Conversely, the dry season (which lasts from August to November) is characterised by air masses arriving from the south east, predominately travelling over urban and agricultural areas of Brazil. As a result, they often bring anthropogenic emissions of trace gases and associated aerosols to the ATTO site, leading to elevated concentrations of species such as black carbon and carbon monoxide (Saturno et al., 2018a). Both seasons

are also affected by long-range transport from Africa (Holanda et al., 2020; Wang et al., 2016).

In addition to a base camp, electrical installations and various container units that house instruments, the site is composed of three measurement towers: an 80-m mast used for aerosol measurements; an 80-m walk-up tower (2° 08.637′S, 58° 59.992′W,





m a.s.l) which can accommodate larger instrumentation; and a 325-m tower (2° 08.602′S, 59° 00.003′W, 120 m a.sl.), on which instruments for long-term measurements are installed. The GRAEGOR system for this campaign was installed on the

80-m walk-up tower.

For the consideration of flux fetch distance, wherein accurate measures of fluxes for a surface are limited by the homogenous extent of the surface's roughness elements, a flux footprint and thus fetch requirement of 5.2 km was calculated based on the geometric mean of the sample heights, and from the formulation given by Monteith and Unsworth (2013). Consequently, the fetch distance lies within the region of *terra firme* forest which extends 5.5 km in all directions from the tower.

**2.2    Instrumentation**

### 2.2.1    Gradient of Aerosols and Gases Online Registration (GRAEGOR)

The GRAEGOR (ECN, The Netherlands) is a semi-autonomous, wet chemistry instrument capable of online quantification of the concentrations of the water-soluble inorganic trace gases $NH_3$, $HCl$, $HONO$, $HNO_3$ and $SO_2$, and their associated aerosol counterparts $NH_4^+$, $Cl^-$, $NO_2^-$, $NO_3^-$ and $SO_4^{2-}$, at hourly resolution at two separate heights (Thomas et al., 2009). It consists

of two sample boxes and a detector box at ground level. For this study, the sample boxes were set at two heights on the 80-m walk up tower: $z_1 = 42$ m and $z_2 = 60$ m.

Each sample box consists of a horizontally-aligned wet rotating annular denuder (WRD) (Keuken et al., 1988) and a steam jet aerosol collector (SJAC) (Slanina et al., 2001) connected in series. Air is simultaneously drawn through both sample boxes at a rate of 16.7 $L\,min^{-1}$, kept constant through critical orifices located downstream of the SJACs. The inlets of the sample

boxes are directly connected to the WRDs via a 0.3 m length high-density polyethylene (HDPE) tubing, which minimises losses of $HNO_3$ and $NH_3$. A HDPE insect gauze is attached to the filters, preventing insects or coarse debris entering the filter. The air streams first pass through the WRDs, which are coated in a continuously replenishing sorption solution of 18.2 $M\Omega$ double deionized (DDI) water. Water-soluble trace gases contained within the laminar air flows diffuse into the liquid sorption solution, which is then fed to the detector box at ground level for analysis. Free of trace gases, the air streams then enter the

SJACs and is mixed with water vapour fed from the DDI solution. This precipitates a supersaturation event, such that any particles contained in the air streams rapidly (0.1 s) grow to droplets of 2 μm diameter. The particle-containing droplets are then separated from the air steams by use of a cyclone, and are fed as liquid samples to the detector box. To prevent biological contamination of the WRDs, the DDI solution includes 0.6 mL of 30% hydrogen peroxide ($H_2O_2$) (9.8 M) per 10 L of DDI.

A series of liquid-pressure regulators were placed in the path of the liquid samples being fed to the detector box in order

to prevent damage to it caused by the high hydrostatic pressures in the 42 m and 60 m high sample columns. Liquid samples from the SJACs and WRDs are analysed for $NH_4^+$ and $NH_3$ respectively by a flow injection analysis (FIA) unit (Norman et al., 2009; Wyers et al., 1993). A 761 compact ion chromatography (IC) unit (Metrohm, Switzerland), equipped with a Dionex AS12 column, determines the liquid concentrations of $HCl/Cl^-$, $HONO/NO_2^-$, $HNO_3/NO_3^-$ and $SO_2/SO_4^{2-}$ in the WRD and SJAC liquid streams respectively based on the measured anion conductivity of the samples compared to a 50 ppb $Br^-$ reference

standard added to the sample solution, taking into account the specific conductivities of the various ions compared with $Br^-$.



A flow control scheme enables continuous analysis of liquid samples. Air concentrations relative to moist air, reported as mass concentrations at ambient temperature and pressure, are derived from the measured liquid concentrations according to:

$$c_i = c_{\text{liq}} \cdot \frac{(Q_{sample} + Q_{\text{Br}})}{Q_{sample}} \cdot \frac{c_{\text{IS}}}{c_{\text{Br}}} \cdot \frac{Q_{\text{sample}}}{Q_{\text{air}}} \cdot \frac{Mw_{i(\text{air})}}{Mw_{i(\text{liq})}} \qquad (1)$$

where $c_{\text{liq}}$ is the liquid concentration of the species measured; $Q_{sample}$, $Q_{\text{Br}}$ and $Q_{air}$ are the sample, internal Br$^-$ standard, and air mass flow rates, respectively; and $c_{\text{IS}}$ and $c_{\text{Br}}$ are the expected internal standard concentration and the detected concentration of the internal standard. The ratio of the molecular weights for air ($Mw_{i(\text{air})}$) and liquid ($Mw_{i(\text{liq})}$) are included to account for the mass differences between the measured ions in the liquid sample and the corresponding gas phase species. For aerosol species, this ratio is equal to 1. The GRAEGOR therefore provides a half hourly averaged measurement of trace gas and aerosol concentrations for each height and species.

The concentrations of the trace gases and aerosols measured by the GRAEGOR are expressed in terms of mass per volume in units of µg m$^{-3}$ at ambient temperature and pressure. Equivalent ambient molar mixing ratios ($r_i$), with respect to moist air, were calculated using the following formulation

$$r_i = \frac{RT}{pM_i} \times c_i \qquad (2)$$

where $R$ is the gas constant (8.314 J K$^{-1}$ mol$^{-1}$), $M_i$ is the molecular weight of the trace gas or aerosol, $c_i$ is the concen-
tration in µg m$^{-3}$ of the trace gas or aerosol, $p$ is the air pressure in Pa and $T$ is the temperature in K.

Calibration of the FIA unit is autonomous, conducted 24 hours after the GRAEGOR begins measurement after start up and every 72 hours afterwards. The calibration uses three liquid NH$_4^+$ samples of 0, 50, and 500 ppb concentration. For this study, a total of 10 autonomous internal calibrations took place. The IC unit is continuously calibrated by the addition of the 50 ppb Br$^-$ internal standard which is added to every liquid IC sample.

Sample box airflows were monitored continuously via the pressure drop across a flow restrictor, calibrated every five days using a Model 4140 Mass Flowmeter (TSI, USA) measuring at ambient volumes (L min$^{-1}$). Additional checks of the instrument performance were conducted daily, for example visual checks that the WRDs or SJACs were not contaminated.

Due to the short inlet length and absence of any size selection, measurements of aerosol taken by the GRAEGOR are of water-soluble total suspended particulate (TSP). Furthermore, as the instrument measures any compound that dissociates to
form the measured anion, the GRAEGOR has a number of potential artefacts. These include interferences in HONO measurements from NO$_2$ during periods of high SO$_2$ concentrations (discussed in detail in 4.3) (Spindler et al., 2003) and interference in HNO$_3$ measurements at night from dinitrogen pentoxide (N$_2$O$_5$). Nevertheless, the GRAEGOR has proven capable of time-resolved flux measurements in previous campaigns (Ramsay et al., 2018; Twigg et al., 2011; Wolff et al., 2010b).



### 2.2.2 Supplementary Measurements

The ATTO site is equipped with an extensive suite of other instruments that provide long-term observations of meteorology, gases and particle properties. Wind speed, wind direction, sensible heat ($H$), air pressure ($p$), and frictional velocity ($u_*$) were measured at 46 m on the 80-m walk-up tower using an ultrasonic anemometer (Gill WindMaster). Continuous measurements of relative humidity and air temperature (both measured using a Vaisala HMP45C-L), rainfall (HS Hyquist TB4-L rain gauge), and net radiation (Kipp and Zonnen Net Radiometer) were also available. Concentrations of equivalent black carbon ($BC_e$)

were measured by an aethalometer (Magee Scientific AE33) at 325 m on the ATTO Tall Tower, and concentrations of carbon monoxide (CO) were measured at 52 m by a Picarro CKADS18. Also presented in this study are concentrations of $NH_4^+$, $Cl^-$, $NO_3^-$ and $SO_4^{2-}$ recorded by a Time-of-Flight Aerosol Chemical Species Monitor (ToF-ACSM, Aerodyne Inc.) at 321 m on the ATTO Tall Tower.

### 2.3 Micrometeorology

#### 2.3.1 Modified Aerodynamic Gradient Method

The aerodynamic gradient method (AGM) is based upon flux-gradient similarity theory, which assumes that the flux of a tracer $c$ (such as a gas or particle) can be determined if its vertical concentration gradient and its diffusion coefficient are known (Foken, 2008). In this study, a modified hybrid form of the AGM is used, whereby the flux of a trace gas or aerosol species can be determined from the vertical concentration difference of the species ($\Delta_c$), and a series of stability parameters and the

friction velocity ($u_*$) derived by eddy-covariance from fast-response ultrasonic anemometry (Flechard, 1998):

$$F_c = -u_*\kappa \frac{\Delta_c}{\ln\left(\frac{z_2-d}{z_1-d}\right) - \Psi_H\left(\frac{z_2-d}{L}\right) + \Psi_H\left(\frac{z_1-d}{L}\right)} \tag{3}$$

Here, $\kappa$ is the dimensionless von Kármán constant ($\kappa = 0.41$); $z_2$ and $z_1$ are the heights at which the concentrations were measured (60 m and 42 m, respectively, in this study); $d$ is the zero-plane displacement height in m; $\Psi_H$ is the integrated form of the heat stability correction term, included to account for deviations from the log-linear profile; and $\zeta = (z-d)/L$, is a

dimensionless atmospheric stability parameter based on $L$, the Obukhov length. By convention, a negative flux value denotes deposition to the surface while a positive flux an emission from the surface.

The zero-plane displacement height, $d$, is a critical parameter for calculation of the flux, and for a closed canopy is related to the canopy height, $h_c$ ($d = 0.66$ to $0.9 \times h_c$). The analysis of this campaign uses a value of $d = 33.4$ m as determined by Chor et al. (2017) from measurements of the logarithmic wind profile at the same tower.



### 2.3.2 Calculation of dry deposition velocities

The dry deposition velocity ($V_d$) is the negative ratio of the flux of the species to its concentration at a reference height ($z$) with consideration to the zero-plane displacement height:

$$V_d(z-d) = -\frac{F_c}{c_z(z-d)} \tag{4}$$

For gases, the deposition velocity can also be determined from the resistance analogy for dry deposition (Fowler and Unsworth, 1979; Wesely et al., 1985). Here, $V_d$ is the reciprocal of the sum of the aerodynamic resistance $R_a$, the quasi-laminar boundary layer resistance $R_b$, and the canopy resistance $R_c$:

$$V_d(z-d) = \frac{1}{R_a(z-d) + R_b + R_c} \tag{5}$$

$R_a$ and $R_b$ can be calculated from 6 and 7 (Garland, 1977):

$$R_a(z-d) = \frac{u(z-d)}{u_*^2} - \frac{\Psi_H(\zeta) - \Psi_M(\zeta)}{\kappa u_*} \tag{6}$$

$$R_b = (Bu_*)^{-1} \tag{7}$$

where $\Psi_M$ is the integrated form of the momentum stability correction term; and $B$ is the sub-layer Stanton number (Foken, 2008), the product of the turbulent Reynolds number and the Schmidt number.

If the $V_d$ of a trace gas is known from its flux via Eq. 4, and $R_a$ and $R_b$ are calculated using micrometeorological data, the canopy resistance $R_c$ can be inferred from rearranging Eq. 5. Similarly, a theoretical maximum deposition velocity ($V_{max}$) for a trace gas can be determined if $R_a$ and $R_b$ are known, by setting $R_c = 0$, which is equivalent to assuming perfect absorption of the gas by the canopy:

$$V_{\max}(z-d) = \frac{1}{R_a(z-d) + R_b} \tag{8}$$

The deposition of particles is more difficult to parameterise using the dry deposition resistance analogy, due to the different behaviour of particles compared to gases. In particular, the physical transport of particles through the quasi-laminar boundary layer is dependent on processes other than Brownian diffusion, such as impaction and interception. Consequently, although aerosol deposition velocities can be calculated as per Eq. 4, the associated theoretical $V_{max}$— which depends on measurements of $R_b$—cannot. Furthermore, due to the complexity in modelling the deposition process for larger particles, the deposition





velocity for a particle is often replaced by an associated surface deposition velocity ($V_{ds}$) value, parameterised by (Wesely et al., 1985):

$$230 \quad V_{ds} = \frac{1}{\left( \frac{1}{V_d} - R_a \right)} \tag{9}$$

### 2.3.3 Correction factors for AGM in roughness sub-layer

The aerodynamic gradient method is ultimately based on Monin-Obukhov similarity theory (MOST). One of its assumptions is that fluxes are measured in the inertial sub-layer, where fluxes deviate little with height. For this reason, the inertial sub-layer is often termed the "constant flux layer" (CFL). However, in the roughness sub-layer (RSL) which extends over the individual roughness elements of the surface, MOST does not strictly hold (Garratt, 1980). As a result, one of the underlying assumptions of the AGM is invalid, and consequently flux measurements using AGM can be erroneous (De Ridder, 2010).

Over forests, the roughness sub-layer can extend to almost three times the height of the canopy. Indeed, it is virtually impossible to make gradient flux measurements that avoid measuring within the roughness sublayer, both for logistical reasons, but also because gradients become increasingly weak at higher height and because of the limitations of the CFL (Dias-Júnior et al., 2019). As with other studies, the flux measurements presented here were made at least partially within the RSL of the rainforest, where the height of the canopy was 37.1 m and the roughness sublayer height therefore extended to an estimated 111 m.

As the profiles of concentrations and turbulence deviate from the logarithmic shape assumed by Eq. 3 within the RSL, fluxes calculated with the standard approach are likely to be underestimated compared to the true flux value (Raupach and Legg, 1984). However, the overall flux gradient relationship within the roughness sublayer can still hold (Simpson et al., 1998) and be used to determine fluxes, but correction factors (also termed enhancement factors) must be implemented to account for measuring within the roughness sub-layer.

Work by Chor et al. (2017) at the ATTO site has led to development of such a correction factor, hereafter termed $\gamma_F$, that can be applied to flux measurements made using AGM above tropical rainforest. The $\gamma_F$ value is dependent upon atmospheric stability, with a larger correction factor applied during stable atmospheric conditions compared to unstable conditions. This reflects the findings made by Zahn et al. (2016) over tropical rainforest, that the solar zenith angle alters the predictions of scalars by MOST in the roughness sub-layer, with best agreement between observations and predictions at noon. Using measurements of $L$ as a parameter for stable and unstable atmospheric stability, the values of $\gamma_F$ developed by Chor et al. (2017) were applied to AGM flux calculations throughout this study, after it was verified that they provide good agreement between measured and theoretically derived deposition velocities for HCl and $HNO_3$ (see3.3.1).



### 2.4 Estimation of errors

#### 2.4.1 GRAEGOR Limits of Detection (LOD)

The concentration limit of detection (LOD) (defined as $3\sigma$ above the background signal, where $\sigma$ is the standard deviation) is of critical importance when measuring in regions of very low concentrations such as the Amazon rainforest. The LOD for each
species measured by the GRAEGOR was determined from a field blank test, which was conducted during the campaign over a 22 h period from 18:00 local time on 23 to 16:00 local time on 24 October 2017. As detailed by Thomas et al. (2009), the field blank test to determine concentration LODs involves switching off the sample box air pump and sealing the air inlets of the samples boxes, while leaving the rest of the system operating under measurement conditions. LODs are then determined as $3\sigma$ from the resulting background signal. Concentration LODs determined during this campaign are presented in Table 1 for
individual trace gas and associated aerosol species, respectively.

#### 2.4.2 Error in concentration measurements

The overall error in concentration measurements ($\sigma_m$ for the trace gases and aerosol components can be expressed as the product of the mixing ratio ($m$) with the individual error measurements, estimated by using a Gaussian Error Propagation approach (Trebs et al., 2004):

$$\sigma_m = m\sqrt{\left(\frac{\sigma_{m_{liq}}}{m_{liq}}\right)^2 + \left(\frac{\sigma_{Br_{(std)}}}{Br_{(std)}}\right)^2 + \left(\frac{\sigma_{Q_{Br}}}{Q_{Br}}\right)^2 + \left(\frac{\sigma_{m_{Br}}}{m_{Br}}\right)^2 + \left(\frac{\sigma_{Q_{air}}}{Q_{air}}\right)^2} \tag{10}$$

Each term in the propagation product denotes a measurement parameter and its associated standard deviation ($\sigma_\chi$). In order, these are the mixing ratio of the compounds found in the liquid sample ($m_{liq}$), the mixing ratio of the $Br^-$ standard ($Br_{(std)}$), the flow rate of the internal $Br^-$ standard ($Q_{Br}$), the mixing ratio (as analysed by the IC system) of the $Br^-$ standard ($m_{Br}$) and the air mass flow through the system ($Q_{air}$). This formulation applies strictly for calculating the error in concentration
measurement of species measured using IC. For $NH_3$ and $NH_4^+$, which were analysed using FIA, the error in concentration measurement can also be determined by using Eq. 10 and omitting the terms for $Br_{(std)}$ and $m_{(Br)}$ and replacing the factor $Q_{Br}$ with $Q_S$, the flow rate of the $NH_3$/$NH_4^+$ liquid sample. Calculated uncertainties ranged from 9%–19%, with $Q_S$, $Q_{Br}$, and $m_{Br}$ the largest contributors to total measurement uncertainty.

#### 2.4.3 Error in flux measurements

As outlined by Wolff et al. (2010b) and Ramsay et al. (2018), the flux measurement error ($\sigma_F$) for a trace gas or aerosol is composed of two terms, the product of the error in the concentration difference ($\Delta_c$) and its associated standard deviation





$(\sigma_{\Delta_c})$ with the error in the flux-gradient relationship (here, expressed as a transfer velocity), which is dominated by the error in $u_*$ ($\sigma_{u_*}$); and the flux ($F$) of the trace gas or aerosol measured:

$$\sigma_F = F\sqrt{\left(\frac{\sigma_{u_*}}{u_*}\right)^2 + \left(\frac{\sigma_{\Delta_c}}{\Delta_c}\right)^2} \tag{11}$$

The error in the concentration difference can be determined through extended side-by-side measurements, where both sample boxes are placed at the same height and are supplied with a common air inlet. The instrument is then allowed to operate normally. The concentrations measured by both sampling boxes during this side-by-side sampling period are plotted against each other and fit with orthogonal regression. Using the orthogonal fit equation, the concentrations for the side-by-side sampling period and the wider campaign can then be corrected to account for systematic errors between each sample box. After
correction, the remaining scatter in the side-by-side sampling concentrations (the residuals) is used to determine the error in the concentration difference. For the ATTO campaign, extended side-by-side measurements were conducted on 6 November at the end of the measurement period, with both sample boxes placed at 60 m.

    The value of $\sigma_{u_*}$ is dependent upon the sonic anemometer used to measure $u_*$ and the atmospheric stability at the time of measurement (Foken, 2008; Nemitz et al., 2009). For this campaign, a value of 10% for $\sigma_{u_*}$ was used during non-neutral
conditions, and 12% for neutral conditions.

    The median error values in flux calculations, as a percentage of flux values, is presented for trace gases and aerosol components in Tables 2 and 3, respectively. These values are in line with those calculated for previous studies (Ramsay et al., 2018; Thomas et al., 2009; Wolff et al., 2010a).

## 3   Results

### 3.1   Meteorology and indicators of pollution

Figure 1 presents hourly time series of the net radiation, rainfall, relative humidity, air temperature, wind direction and wind speed measured during the campaign. Also presented are the mass concentration of black carbon ($M_{BC_e}$) and mixing ratio of carbon monoxide ($c_{CO}$). The values of $M_{BC_e}$ and $c_{CO}$ have been used in previous studies at ATTO to demarcate periods of near-pristine and polluted conditions. Thus Pöhlker et al. (2018) defined "pristine rainforest" ($PR$) conditions as periods when
$M_{BC_e}$ values are $<0.01\ \mathrm{\mu g\,m^{-3}}$ for over 6 hours. Alternatively, or in combination with $M_{BC_e}$, periods when $c_{CO}$ values are below the monthly background CO concentrations recorded at the Ascension Island hemispheric background reference station (https://www.esrl.noaa.gov/gmd/dv/site/?stacode=ASC; last access: 22 December 2019) are also considered $PR$ conditions. During this campaign, there were no recorded periods when $M_{BC_e}$ or $c_{CO}$ met these criteria, and therefore no period of $PR$ conditions. This is typical for dry season conditions (Pöhlker et al., 2016).
While $PR$ conditions (according to the above definition) were not observed, there were periods when $M_{BC_e}$ over a 6 h period was close to falling below $0.01\ \mathrm{\mu g\,m^{-3}}$. For example, between 12:00 on 8 October and 09:00 on 9 October, $M_{BC_e}$





values varied between 0.01 and 0.02 $\mu g\,m^{-3}$. Periods where $M_{BC_e}$ values approach the $PR$ criterion were associated with periods of rainfall and north to north-easterly winds. For the remainder of this paper, periods when the values of $M_{BC_e}$ and $c_{CO}$ approached conditions for $PR$ status (0.01 $\mu g\,m^{-3}$ and 150 ppb, respectively, over 6 hours) are termed "near-$PR$" conditions.

Conversely, there are periods where $M_{BC_e}$ and $c_{CO}$ values notably exceeded their mean values (0.04 $\mu g\,m^{-3}$ and 280 ppb respectively), for example the period between 21 and the 25 October (Figure 1). During this time, values of $M_{BC_e}$ increase steadily from 0.04 $\mu g\,m^{-3}$ to a maximum of 0.12 $\mu g\,m^{-3}$ at 00:00 on 25 October. A sharp decrease in $M_{BC_e}$ occurs at 04:00 on the same day, coinciding with a period of precipitation, the first since 18 October. This 5-day period is also noted for comparatively drier, warmer conditions and a prevailing wind direction from the east to south-east. Periods where there was a 320  6 h exceedance of the mean value of $M_{BC_e}$ (0.04 $\mu g\,m^{-3}$) with associated drier, warmer conditions are referred to hereafter as "polluted" conditions.

### 3.2   Concentrations of inorganic trace gases and associated aerosol counterparts

Summary statistics for the inorganic trace gases and associated aerosol counterparts measured at 60 m are presented in Table 1. The table also includes the associated limit of detection values. The times series of inorganic trace gas concentrations, in $\mu g\,m^{-3}$ 325  and ppb, at 42 m and 60 m are shown in Figure 2, and the corresponding time series of associated aerosol concentrations are shown in Figure 3. For comparison, Figure 3 also presents the concentrations of particulate $NH_4^+$, $Cl^-$, $NO_3^-$ and $SO_4^{2-}$ measured by the ToF-ACSM taken at 321 m on the Amazon Tall Tower. Gaps in the GRAEGOR time series are due to automated calibrations of the instrument, instrument failure, or periods where liquid or air flow were unstable.

    Table 1 shows that the mean and median concentrations of all trace gases and associated aerosol species exceeded their 330  limit of detection except for nitrite ($NO_2^-$). Particulate $NO_2^-$ is particularly difficult to quantify using wet chemistry methods owing to its low ambient concentrations. Previous attempts to measure $NO_2^-$ using the GRAEGOR at rural sites have also been unsuccessful (Ramsay et al., 2018; Wolff et al., 2010b). Consequently, $NO_2^-$ data are not discussed further in this paper.

    All aerosol species (with the exception of $NO_2^-$) had mean and median concentrations greater than the associated inorganic trace gases. This was the case at both measurement heights. For example, the mean and median concentration values of $NH_4^+$ 335  at 42 m (0.30 $\mu g\,m^{-3}$ and 0.28 $\mu g\,m^{-3}$ respectively) exceeded those recorded for $NH_3$ at the same height (0.27 and 0.22 $\mu g\,m^{-3}$). The difference is most pronounced between $NO_3^-$ and $HNO_3$, and $SO_4^{2-}$ and $SO_2$, with a mean value of 0.47 $\mu g\,m^{-3}$ for $NO_3^-$ at 60 m compared to a corresponding mean value of 0.25 $\mu g\,m^{-3}$ at the same height for $HNO_3$; and a mean value of 0.51 $\mu g\,m^{-3}$ for $SO_4^{2-}$ at 60 m compared to a mean value of 0.23 $\mu g\,m^{-3}$ for $SO_2$ at the same height. The predominance of aerosol phase over gas phase for these species has been noted at other rural and forest sites; for example Wolff et al. (2010b) 340  reported median $NO_3^-$ and $HNO)3$ concentrations of 0.48 $\mu g\,m^{-3}$ and 0.12 $\mu g\,m^{-3}$ using the GRAEGOR above a rural forest in SE Germany.

    Concentrations varied between near-$PR$ and polluted periods. Minimum values for all aerosol and gas species—which fall below their respective instrumental LODs—occurred during near-$PR$ conditions. Conversely, the maximum concentration values recorded for all species occurred during the longest polluted period of the campaign (21–25 October). In particular, 345  $Cl^-$ and $NO_3^-$ reach their respective maximum concentrations of 1.35 $\mu g\,m^{-3}$ and 2.07 $\mu g\,m^{-3}$ at 23:00 on 21 October.



Concentrations of $NH_3$ and $HNO_3$ increase from 21 October to reach maximum values of 1.94 µg m$^{-3}$ and 1.04 µg m$^{-3}$, respectively, at noon on 23 October.

The extent of agreement in aerosol concentrations between the GRAEGOR at 60 m and the ToF-ACSM at 321 m depends on the species (Figure 3). Measurements of $SO_4^{2-}$ are in best agreement. Linear regression analysis for the full campaign showed

a near 1:1 agreement between $SO_4^{2-}$ measured by GRAEGOR and ToF-ACSM ($m = 0.89$, $R^2 = 0.45$). During the period from 18 to 26 October, agreement was particularly good ($m = 0.97$, $R^2 = 0.65$). Similarly, although not as statistically robust as for the $SO_4^{2-}$ measurements, there is near-linear relationship between $NH_4^+$ concentrations measured by GRAEGOR at 60 m and ToF-ACSM at 321 m ($m = 0.85$, $R^2 = 0.35$).

In contrast, there are significant differences between GRAEGOR and ToF-ACSM measurements for both $NO_3^-$ and $Cl^-$.

While there is some agreement in overall trends between GRAEGOR and ToF-ACSM measurements of $NO_3^-$, with both instruments recording a maximum in $NO_3^-$ at 23:00 on 21 October 2018 (ToF-ACSM = 0.54 µg m$^{-3}$, GRAEGOR, 60 m = 2.07 µg m$^{-3}$), in general the GRAEGOR measurements of $NO_3^-$ are a factor of 3–4 larger than those from the ToF-ACSM. The difference in $Cl^-$ concentration is even more pronounced. The median concentration for $Cl^-$ from the ToF-ACSM is 0.02 µg m$^{-3}$ whilst the median value from the GRAEGOR at 60 m is 0.14 µg m$^{-3}$. 93% of the GRAEGOR $Cl^-$ measurements are

above its LOD of 15 ng m$^{-3}$ .

The median (0.06 µg m$^{-3}$) and mean (0.07 µg m$^{-3}$) values for the inorganic trace gas nitrous acid (HONO) remained above the detection limit of the instrument (30 ng m$^{-3}$) at both sampling heights. Although the diel cycle of HONO exhibited a maximum during night and a minimum during the day (0.02 µg m$^{-3}$ at 14:00), it remained above the detection limit even during daylight hours (Figure 4), which, given the high photolysis rate of HONO during daytime, implies the presence of a

daytime source. Similarly, median diel $SO_2$ concentrations remained above the LOD throughout the campaign. $SO_2$ is usually considered a marker for anthropogenic emissions, but its presence at concentrations above detectable limits during near-$PR$ conditions might be at least in part supported by biogenic sources. Previous measurements had found $SO_2$ concentrations close to the lowest values observed in this study and had attributed them partly to biogenic emissions (Andreae et al., 1990a; Andreae and Andreae, 1988). There are also periods when the trace gas HCl—another marker of anthropogenic emissions, originating

from combustion activities and the reaction of seasalt with $HNO_3$— is recorded at elevated concentrations above its detection limit.

### 3.3 Fluxes, Deposition Velocities, and Canopy Resistances

#### 3.3.1 Fluxes of inorganic trace gases

Figure 5 shows the average diel cycles of the deposition velocities in comparison with those of $V_{max}$ for HCl and $HNO_3$.

Two sets of values are presented: values calculated using the standard modified aerodynamic gradient method (Section 2.3.1) without the application of a correction factor for measuring within the roughness sublayer, termed "pre-correction values"; and values calculated with the application of a flux correction factor developed by Chor et al. (2017), $\gamma_F$, discussed in Section 2.3.3, which adjusts values derived from the aerodynamic gradient method when measuring in the roughness sublayer, termed





"post-correction values". Due to their high water solubility (and resulting large effective Henry coefficient), HCl and $HNO_3$

are expected to deposit at $V_{max}$ (Lelieveld and Crutzen, 1991), unless chemical conversions affect their fluxes (Nemitz et al., 2000; Twigg et al., 2011). The correction brings the $V_d$ for these gases in close agreement with $V_{max}$, within the measurement error. The correction increases the average $V_d$ of $HNO_3$ from 10.2 to $12.4 \, \mathrm{mm \, s^{-1}}$ (average $V_{max} = 12.3 \, \mathrm{mm \, s^{-1}}$) and that of HCl from 12.5 to $15.2 \, \mathrm{mm \, s^{-1}}$ (average $V_{max} = 15.3 \, \mathrm{mm \, s^{-1}}$). This suggests that, overall, the $\gamma_F$ correction works well, and the remainder of the paper discusses post-correction values only. With this consideration in mind, Figure 6 shows the average

diurnal cycles of the post-$\gamma_F$ corrected deposition velocity in comparison with that of $V_{max}$ for the remaining trace gases measured: $NH_3$, HONO and $SO_2$ .

Table 2 presents a statistical summary of the calculations for fluxes, deposition velocities ($V_d$), theoretical maximum deposition velocities ($V_{max}$) and canopy resistances ($R_c$) for the inorganic trace gases measured during the campaign. As discussed above, with the roughness sublayer correction of Chor et al. (2017), both HCl and $HNO_3$ are observed to deposit at $V_{max}$

within the error of the measurement, with a canopy resistance $< 3 \, \mathrm{s \, m^{-1}}$, although the results would be sensitive to the $R_b$ parameterisation used, which for forests can vary significantly.

Time series for the post-filtered fluxes of the inorganic trace gases measured are shown in Figure 7. The inorganic trace gases $HNO_3$, $SO_2$ and HCl were nearly always deposited to the surface. Any upward fluxes calculated for these gases lay within their respective error ranges. Fluxes which exceed the median values for these gases, and the maximum calculated fluxes for

these species, were recorded during the drier, warmer "polluted" conditions that prevailed from 18 to 26 October 2017. For example, the maximum calculated flux for $SO_2$, and the largest flux of any species measured during the campaign, was -33 $\mathrm{ng \, m^{-2} \, s^{-1}}$ which occurred on 21 October at 11:00. Conversely, while increased deposition fluxes are observed for $NH_3$ and HONO during this same period, multiple periods of emission were recorded for these gases throughout the campaign. Although the predominant pattern of surface-atmosphere exchange throughout the campaign for HONO and $NH_3$ was deposition to

the surface, as reflected in their respective median flux and $V_d$ values, periods of emission are a significant proportion of overall surface-atmosphere exchange. For HONO and $NH_3$, respectively, 26% and 19% of calculated fluxes were positive, i.e. emissions. The median diel pattern of trace gas emissions is highlighted in Figure 8. HONO emissions were concentrated in the early morning, with positive median values indicating a prevalent pattern of emission present at 07:00 and 08:00. In contrast, $NH_3$ emissions were observed in the afternoon, from 14:00 to 16:00 hours. The other trace gases—HCl, $HNO_3$ and

$SO_2$ – showed maximum deposition fluxes in the afternoon, with decreased fluxes during the night and early morning hours.

### 3.3.2 Fluxes of associated ionic aerosol counterparts

A statistical summary of fluxes and deposition velocities for the aerosol species is presented in Table 3. Also included for each species is the minimum detectable flux ($F_{LOD}$), and the percentage of calculated fluxes which exceed this value ($f_{LOD}$ %).

Median $V_d$ values for $NH_4^+$ and $SO_4^{2-}$ were 2.64 and $2.81 \, \mathrm{mm \, s^{-1}}$ respectively. In the comparison of GRAEGOR and

ToF-ACSM concentration measurements outlined in 3.2, we found a reasonable agreement for $NH_4^+$ and $SO_4^{2-}$, considering the difference in measurement height and instrumentation. Given that the ToF-ACSM measures only the sub-micron ($<1 \, \mu m$ particle diameter) range, this suggests that the $NH_4^+$ and $SO_4^{2-}$ quantified by the GRAEGOR were also dominated by the





sub-micron range. From process-orientated modelling of aerosol $V_d$, it has been suggested that particle $V_d$ increases over increasingly rough surfaces. In a meta-analysis of field flux data, Gallagher et al. (2002) parameterised this relationship as a

function of the surface deposition velocity, $V_{ds}$, and the surface roughness (given as the surface roughness length, $z_0$, in m):

$$V_{ds} = 0.581 \log (z_0) + 1.86 \qquad (12)$$

Using the median value of the surface roughness lengths calculated at the site (and including only lengths with a valid calculated value of aerosol $V_d$) yields a value of 2.86 m for $z_0$. Substituting this into the Eq. 12 parameterisation suggests a $V_{ds}$ of 2.1 mm s$^{-1}$ for sub-micron particles. Values of NH$_4^+$ and SO$_4^{2-}$ $V_d$ converted to $V_{ds}$ using Eq. 9 results in a median

$V_{ds}$ value for NH$_4^+$ and SO$_4^{2-}$ of 2.9 and 3.3 mm s$^{-1}$, respectively. Although these values are higher than the parameterised value, Eq. 12 was derived specifically for particles in the range 0.1–0.2 µm. Larger particle sizes, would have higher $V_{ds}$ for a given value of $u_*$ (Davidson et al., 1982; Slinn, 1982). Thus, if the particle size range for NH$_4^+$ and SO$_4^{2-}$ exceeds 0.2 µm, but remains in the sub-micron range, the measured median $V_d$ would exceed the parametrised value.

In contrast to $V_{ds}$ values for NH$_4^+$ and SO$_4^{2-}$, which are in the range for parameterised values for the site, the median $V_{ds}$

values for Cl$^-$ and NO$_3^-$ are 3 to 4 times greater than the parametrised value of 2.1 mm s$^{-1}$. The median $V_{ds}$ value for Cl$^-$ is 10.2 mm s$^{-1}$, while for NO$_3^-$ it is 7.6 mm s$^{-1}$. As the parameterised value holds only for particle diameters between 0.1–0.2 µm, and considering that modelling indicates an increase in $V_{ds}$ with increasing particle size, the larger median $V_{ds}$ values for Cl$^-$ and NO$_3^-$ are consistent with the GRAEGOR vs ACSM comparison which suggests that these aerosol counterparts were present in the super-micron ($>$ PM$_1$) fraction.

A time series of the aerosol counterpart fluxes is presented in Figure 9. The predominant direction of surface-atmosphere exchange for all aerosol species was deposition, as reflected in the median flux values in Table 3. However, individual emission fluxes were recorded for all species, with the maximum emission values for Cl$^-$ and SO$_4^{2-}$ (+3.6 ng m$^{-2}$ s$^{-1}$ and +4.3 ng m$^{-2}$ s$^{-1}$, respectively) being particularly large. The time series of values is filtered for identifiable errors in measurement and for micrometeorological values that fall outside specified limits (Section 2.4.3). These emission fluxes are therefore un-

likely to be caused by instrumentation faults or calculation errors. They are, however, limited in duration and overall extent – positive particle emissions are never observed consecutively, occurring exclusively within one-hour periods, and constitute only between $<$1% (NH$_4^+$) to 5% (Cl$^-$) of total fluxes. While particle emission fluxes have previously been observed with the GRAEGOR (Nemitz et al., 2004; Twigg et al., 2011), these previous observations have occurred during periods of known flux divergence.

## 4 Discussion

### 4.1 Long range transport of pollutants – the influence of biomass burning on measurements

All measured gas and aerosol species show significant differences in concentrations between near-pristine and polluted periods (Figures 2 and 3). The minimum recorded concentrations for all species are during periods when $BC_e < 0.02$ µg m$^{-3}$ and $c_{CO}$





$< 150$ ppb. Conversely, maximum concentrations for all species occur between 21 and 25 October 2017, during which time
the concentration of $BC_e$ peaks at $0.14\,\mu g\,m^{-3}$ at midnight on the 25 October along with a peak in $c_{CO}$ of 300 ppb. Calculated
fluxes exhibit the same behaviour, with maximum deposition fluxes occurring during the relatively polluted period. The gases
$NH_3$, HCl and $SO_2$ all have maximum deposition values on 21 October, with a pronounced deposition of $-33\,ng\,m^{-2}\,s^{-1}$ for
$SO_2$ at 11:00 on this day. While $HNO_3$ also shows large deposition fluxes on 21 and 22 October, its maximum deposition value
is on 25 October when the HONO flux is also at its maximum deposition value.

For the relatively polluted period from 21 to 25 October, there is evidently a marked increase in concentrations and fluxes
above the average dry-season background levels. Anthropogenic activity, principally biomass burning, may be the driver for this
increase. This can be assessed from the strength of correlation between trace gases and aerosol concentrations and measured
concentrations of $BC_e$, which acts as a marker for biomass burning and for anthropogenic emissions in general. For all species,
Spearman rank correlation coefficients were statistically significant ($p <0.05$), suggesting a monotonic relation between all
inorganic trace gases and associated aerosols with $BC_e$. Correlations with $BC_e$ were strongest for $NH_3$ ($r_s = 0.60$) and $SO_2$
($r_s = 0.51$), which was also the case for their respective aerosol phases. The weakest correlation between a gas and $BC_e$ was for
HCl ($r_s = 0.29$). HONO and $HNO_3$, while not as strongly correlated with $BC_e$ as $NH_3$ and $SO_2$, showed a moderate positive
correlation. Conversely, there was a weak positive correlation between $NO_3^-$ and $BC_e$, and a very weak positive correlation
for $Cl^-$.

To determine the origin of the polluted air masses arriving at the ATTO site during the relatively polluted period of the
campaign when $BC_e$ concentrations were largest, back trajectory analysis was conducted. Ten-day air-mass back-trajectories
arriving every 3 hours at a height of 500 m a.s.l. between 18 and 25 October 2017 were obtained from the HYSPLIT-4
air trajectory model (Stein et al., 2015) and the Global Data Assimilation System (GDAS) meteorology dataset at $1° \times 1°$
resolution, and analysed using the openair package for R (Carslaw and Ropkins, 2012). The ensemble of back trajectories
per week of the campaign, with associated frequency trajectory plots, is shown in Figure 11. Trajectories arriving during the
third week (20–26 October), when increased concentrations of pollutants were measured, are notable for their origin near the
south-west coast of Africa. They are also differentiated from the other trajectories by the frequency with which they travel
further south over the interior of Brazil, veering sharply to arrive at the site from a southerly direction and thus from over the
populated areas to the east of Manaus. Figure 12 focuses on the path of the daily trajectories grouped by week in the regional
area surrounding the ATTO site, with the location of fires (recorded by the National Aeronautics and Space Administration's
Fire Information for Resource Management Service) overlaid. During the period of increased concentrations from 19 to 24
October, trajectories travel over areas where frequent fires were recorded.

This back-trajectory analysis provides some insight into the origins of the polluted air masses during 21 to 25 October.
During the dry season, a mixture of regional and remote sources contribute to the pollution over the Amazon Basin, with
local sources from deforestation and biomass burning being predominant (Andreae et al., 2012; Pöhlker et al., 2019). Pollution
from the densely populated north-east coast of Brazil adds to the pollution burden throughout the relatively polluted period
(Andreae et al., 2018). In addition to this dry season background pollution, there are periods when long-range transport of
pollutants contributes to the overall pollution burden observed at the ATTO site. The sources for the majority of these LRT





episodes during the dry season are located in southern Africa (Holanda et al., 2020), with volcanic eruptions (Saturno et al.,

2018a) and biomass burning (Pöhlker et al., 2018; Andreae et al., 2018) as two of the attributed causes. As the 10-day back trajectories for 21 to 24 October originate at the west coast of southern Africa, it is likely that the increased concentrations and fluxes of the longer lived aerosols species are due to the long-range transport of biomass burning pollution from southern Africa.

The inorganic gases and aerosol species measured during the ATTO campaign at elevated concentrations during polluted pe-

riods are consistent with signatures of biomass burning; this has been confirmed by investigations into the chemical constituents of smoke from biomass burning in laboratory studies (McMeeking et al., 2009), field studies from atmospheric monitoring stations located near biomass burning point sources (Aurela et al., 2016), and aircraft measurements of plumes from biomass burning (Andreae et al., 2018; Aruffo et al., 2016; Fiedler et al., 2011). Biomass burning is an important source of reactive nitrogen emissions, and emissions of $NH_3$ from biomass burning are the second most important source of global emissions

behind agriculture, accounting for 14% of total terrestrial emissions (Van Damme et al., 2014; Whitburn et al., 2015). The predominant source for the production of $HNO_3$ and HONO in the troposphere is the OH driven oxidation of $NO_2$, which occurs in conditions of elevated $NO_2$ concentrations. In remote areas, where background levels of $NO_2$ are low, the production of $HNO_3$ is limited. However, with injections of anthropogenically derived $NO_2$ into the atmosphere above remote areas, the efficient scavenging of OH by elevated $NO_2$ concentrations leads to the formation, and subsequent deposition, of $HNO_3$

(Mannschreck et al., 2004). Emissions of $NO_x$ from burning during the southern African biomass burning season is a significant contributor to free tropospheric $NO_x$ in the southern hemisphere (Adon et al., 2010; Galanter et al., 2000). Finally, elevated concentrations of $SO_2$ and HCl as well as sub-micron particles such as $SO_4^{2-}$ and $NH_4^+$ have previously been measured in biomass burning plumes, during both ground and aircraft measurements (Burling et al., 2010; Yokelson et al., 2011; Andreae et al., 1998), and the corresponding emission factors have been compiled in Andreae (2019). Adachi et al. (2020) found that

the number fractions of sea salt and mineral dust measured during the Green Ocean Amazon Campaign (February to March 2014) increased three fold during periods when LRT occurred.

The evidence from the correlation and back-trajectory analyses suggests that the presence of $SO_2$ and $NH_3$ (and also of $NH_4^+$ and $SO_4^{2-}$) was primarily driven by biomass burning. For the period from 21 to 24 October, concentrations of $NH_4^+$ and $SO_4^{2-}$ may have been elevated due to biomass burning in the region surrounding the ATTO site, with the possible complement

of plumes from biomass burning originating in southern Africa. Figure 13 highlights this link by presenting concentration-weighted trajectory analyses, which determine the geographic origin for concentration levels of a select species, for $BC_e$, $SO_4^{2-}$ and $NH_4^+$. Areas determined as the source for the highest measured concentration of these three species align with areas in which the most intense (as determined by the fire radiative power of each fire count) biomass burning occurred regionally.

While this holds partly for HONO and $HNO_3$, it only weakly holds for $NO_3^-$, HCl and $Cl^-$). An alternative origin for

these species must therefore be considered and is discussed further in Sections 4.2.3 and 4.3.2 for HCl and for $NO_3^-$ and $Cl^-$ respectively.





### 4.2 Gas-phase concentrations and their controls

#### 4.2.1 Relative contribution of acidic inorganic trace gases to total atmospheric acidity

The relative proportions of inorganic trace gases over the ATTO site during the campaign can give important insight into the overall atmospheric chemistry. As the primary basic gas in the atmosphere, $NH_3$ can react with the acidic gases HCl, $HNO_3$ and $H_2SO_4$ (produced by the oxidation of $SO_2$) to form ammonium salts whose lifetime and behaviour are dependent upon the associated gas. To investigate the importance of the various acidic gases to total acidity at this remote Amazon site, the fractional contribution to total inorganic acid loading for HCl, HONO, $HNO_3$ and $SO_2$ as measured by the GRAEGOR was determined in units of $\mu eq\,m^{-3}$ (Figure 10). Taken as an arithmetic mean value, the fractional contributions of $SO_2$, $HNO_3$ and HCl are similar. While not as significant a contributor in comparison, HONO also contributes at an average fraction of 0.13, which remains consistent throughout the duration of the campaign. The contributions of $SO_2$ and $HNO_3$ average at 0.31 and 0.30, respectively, whilst the contribution of HCl averages at 0.26 but fluctuates throughout the campaign, varying between $\approx$ 0.05–0.10 during near-pristine conditions to almost 0.40 during the polluted period from 19 to 25 October.

#### 4.2.2 Urban Plumes, $NO_x$ and reactive nitrogen formation

Fossil fuel combustion is the primary anthropogenic, and overall predominant, source for $NO_2$ in the troposphere. The increase in $HNO_3$ concentrations on 25 October (also resulting in increased deposition fluxes) could be due to air masses that picked up emissions of $NO_x$ ($NO_2 + NO$) from the urban areas of Manaus and Santarém. Measurements of $NO_2$ downwind and west of the Manaus urban area showed elevated $NO_2$ concentrations in remote areas affected by emission plumes from the city (Kuhn et al., 2010; Trebs et al., 2012; Abou Rafee et al., 2017; Martin et al., 2017). With air masses arriving at the site from the south and south-east, which had travelled over the eastern suburbs of Manaus and the city of Santarém respectively, it is likely that $NO_2$ plumes are responsible for the elevated $HNO_3$ observed on 25 October.

#### 4.2.3 Biogenic drivers of HCl concentrations

While a moderate, positive monotonic relation exists between concentrations of HCl and $BC_e$, it is unlikely that the presence of HCl above the detection limit of the GRAEGOR could be sustained throughout the campaign solely through anthropogenic emissions. HCl is highly reactive and water soluble, with a mean lifetime of $\approx$36 hours (Graedel and Keene, 1995; Kritz and Rancher, 1980). Consequently, it is unlikely that regional or global biomass burning could contribute meaningfully to the HCl concentrations observed at this remote site. The peak in HCl concentrations observed during the relatively polluted periods of the campaign could be a result of biomass burning from local sources in close proximity, but an alternative explanation must be considered for the background concentrations of HCl. Globally, much of the HCl derives from the displacement reaction of $HNO_3$ with aerosol $Cl^-$ compounds; typically with NaCl seasalt, but potentially other $Cl^-$ compounds at this site (see 4.3.2 below). A further potential contributor is oxidation of methyl chloride ($CH_3Cl$), whose predominant natural source is tropical forest (Yokouchi et al., 2002; Xiao et al., 2010). The emissions are driven principally by dipterocarps and ferns (Blei et al.,





2010), whose emission rates are unaffected by abiotic conditions (Yokouchi et al., 2015). Gebhardt et al. (2008) measured an average emission for $CH_3Cl$ of $9.5\ \mu g\,m^{-2}\,hr^{-1}$ over Guyanese and Surinamese rainforest, while Moore et al. (2005) reported

$CH_3Cl$ concentrations above a rainforest canopy in Rondônia, Brazil, confirming that the Amazon rainforest region is a net regional source for $CH_3Cl$.

Sanhueza (2001) proposed an OH driven oxidation pathway for $CH_3Cl$ that terminates with stoichiometric production of HCl. It is thus possible that the tropical forest emissions of $CH_3Cl$, combined with the local high oxidative capacity, could yield the background HCl concentrations observed in this study. However, to confirm this idea, simultaneous measurements

of $CH_3Cl$ and HCl concentrations would be required, together with confirmation of Sanhueza's postulated $CH_3Cl$ oxidation pathway.

### 4.2.4   Anthropogenic and biogenic drivers of $SO_2$ concentrations

This campaign presents the first tower measurements of time-resolved $SO_2$ fluxes over tropical rainforest. Standard commercial $SO_2$ monitors struggle to resolve such low concentrations. Although aircraft (Andreae and Andreae, 1988), denuder tube (Adon

et al., 2013) and filter pack (Paralovo et al., 2019) measurements of $SO_2$ over rainforest exist, they lack the time resolution of the measurements during this campaign or do not measure fluxes. This study has shown that LRT pollution episodes can significantly enhance $SO_2$ deposition fluxes (a maximum deposition flux of $-33.2\ ng\,m^{-2}\,s^{-1}$ was recorded during the most polluted period of the campaign) and that even during relatively pristine conditions, $SO_2$ concentrations remained above the LOD. As Figure 14 demonstrates, the close correlation between $SO_2$ and $BC_e$ suggests that long-term measurements of $SO_2$

over tropical rainforest may be worthwhile as a further method to identify episodes of increased pollution or biomass burning. Long-term measurements would also show whether concentrations of $SO_2$ remain above detection limits during the pristine conditions of the wet season, and help determine potential sources during these periods. It is possible that a biogenic source may have contributed to $SO_2$ measured during the relatively pristine conditions. For example, $SO_2$ could derive from the oxidation by OH of dimethyl sulfide emitted from the rainforest (Jardine et al., 2015).

## 4.3   Aerosol concentrations

### 4.3.1   Aerosol mass fraction – comparison with ACSM

The comparison between ACSM and GRAEGOR water-soluble aerosol concentrations in Section 3.2 indicates good agreement between them for $SO_4^{2-}$ and $NH_4^+$, but significant divergence for $NO_3^-$ and, in particular, $Cl^-$.

Long-term measurements of aerosol chemical composition at the ATTO site using an ACSM have been conducted since

2014, and the first publication of data from 2015 suggested that aerosol chemical speciation varied surprisingly little across the wet and dry seasons (Andreae et al., 2015). As recorded by the ACSM during this campaign, organic aerosols are always the dominant mass fraction (compromising $\approx$70% of aerosol), followed by $SO_4^{2-}$ (10–15%), $BC_e$ (5–11%), $NH_4^+$ ($\approx$5%), $NO_3^-$ ($\approx$4%) and finally $Cl^-$ as the smallest contributor. Focusing only on the aerosol species measured by both the GRAEGOR and ACSM during this dry season campaign, the average ACSM mass fractions are 55% $SO_4^{2-}$, 22% $NH_4^+$, 18% $NO_3^-$ and





5% Cl$^-$. As Figure 15 demonstrates, the total mass fraction contribution to total suspended particulate as measured by the GRAEGOR suggests that the contribution of NO$_3^-$ and Cl$^-$ is more significant than suggested by previous measurements. The relative contribution of each species to TSP as measured by the GRAEGOR in this campaign (in descending order) is: SO$_4^{2-}$ = 34.4%, NO$_3^-$ = 30.8%, NH$_4^+$ = 19.0% and Cl$^-$ = 15.3%. In comparison to ACSM measurements, the relative proportion of SO$_4^{2-}$ is reduced, NO$_3^-$ becomes the second most abundant species with an almost equal contribution to SO$_4^{2-}$, and Cl$^-$—

while remaining the smallest contributor to total mass—has a greater relative contribution to the mass of TSP. Talbot et al. (1990) measured a similar contribution order for the dry season using ion chromatography, with SO$_4^{2-}$ contributing the most to the total mass fraction, and Cl$^-$ the least, but with a differing proportion (SO$_4^{2-}$: 51%, NO$_3^-$: 26%, NH$_4^+$: 19%, and Cl$^-$: 4%). Variations in the ion proportions may be attributable to differences in the number and intensity of long-range transport episodes, which contribute Cl$^-$ and SO$_4^{2-}$, during a given field campaign.

The ACSM samples only the sub-micron (PM$_1$) aerosol size range, while the GRAEGOR samples TSP (<50–100 μm particle diameter). Furthermore, the ACSM only detects non-refractory aerosol compounds, and is therefore insensitive to refractory seasalt and crustal material (Fröhlich et al., 2013). The close similarity in SO$_4^{2-}$ and NH$_4^+$ measurements between the two instruments suggests that the majority of SO$_4^{2-}$ and NH$_4^+$ during the campaign were contained within submicron aerosol and that the SO$_4^{2-}$ represented semi-volatile ammonium compounds. Conversely, the difference between ACSM and

GRAEGOR NO$_3^-$ measurements suggests that most of the NO$_3^-$ was contained within the coarse mode and/or represented non-volatile compounds such as NaNO$_3$ and Ca(NO$_3$)$_2$, and that almost all of the Cl$^-$ measured by the GRAEGOR in this campaign was found in the coarse mode and/or as NaCl. Previous work had found Cl$^-$ to be exclusively associated with the coarse fraction (Talbot et al., 1988, 1990). This is consistent with thermodynamic considerations which would suggest that volatile NH$_4$NO$_3$ aerosol, the NO$_3^-$ compound typically measured by the ACSM, should not exist at the high temperature

and relatively low gas-phase concentrations of NH$_3$ and HNO$_3$ at this site. This was confirmed using the ISORROPIA-2 thermodynamic modelling framework.

### 4.3.2  Potential origins for coarse Cl$^-$ and NO$_3^-$

Consistent with the insensitivity of the ACSM to refractory particles, a possible source for coarse Cl$^-$ aerosols could be seasalt. Although a continental site, intrusions of seasalt through long-range transport have been noted previously at ATTO

(Talbot et al., 1990; Moran-Zuloaga et al., 2018). The presence of sea salt could also account for a source of coarse NO$_3^-$, as the reaction between HNO$_3$ and NaCl would result in the formation of the coarse aerosol NaNO$_3$ (Dasgupta et al., 2007), a refractory aerosol component that would not be detected by the ACSM. The reaction of HNO$_3$ with sea salt would also form HCl, the measured concentrations of which are closely linked to those of Cl$^-$ in this campaign. Alternatively, the strong link between HCl and Cl$^-$ concentrations could be accounted for by biomass burning emissions arriving at the ATTO site, whereby

Cl$^-$ particulate from biomass burning is principally in the form of fine KCl (Pratt et al., 2011). Other crustal material, such as dust and soil particles which are recorded in elevated amounts at ATTO during the dry season (Moran-Zuloaga et al., 2018), could provide a source of coarse NO$_3^-$. These can include a variety of NO$_3^-$-containing mineral species, such as NaNO$_3$,





$Ca(NO_3)_2$ and $Mg(NO_3)_2$ (Karydis et al., 2016). The surface of dust and suspended soil particles could also act as a sink for HCl in the marine boundary layer (Sullivan et al., 2007), allowing the heterogeneous formation of coarse $Cl^-$ particulate.

It has been shown previously that primary biological aerosol particles (PBAPs) contribute the majority of the mass fraction of measured coarse aerosol in the Amazon (Pöschl et al., 2010). The PBAPs over the rainforest consist of a variety of different biological materials, such as plant and animal matter fragments, algae, pollen and fungal spores. The latter contributor is particularly important, as fungi which actively discharge their spores through liquid jets have been identified by Elbert et al. (2007) to be a source of inorganic ions in particulate matter. Fungi that actively discharge their spores do so via a liquid jet,

whereby spores are forcibly discharged from a spore sac (*asci*) along with a liquid mix of sugars and ions, of which $Cl^-$ forms a significant fraction (Trail et al., 2005). The spore itself can rupture under conditions of high relative humidity, resulting in the formation of fragments containing inorganic ions (China et al., 2016). In a chemical imaging analysis of such spore fragments above the Amazon rainforest, China et al. (2018) found that almost 40–60% of these fragments contain $Na^+$ and $Cl^-$ associated as a salt, which appeared "morphologically similar to dry sea salt" and which grew to supermicron sizes in

conditions of high relative humidity. The contribution of fungal spores to total $Na^+$ mass during the wet season over the rainforest was estimated as ≈69% by the same study, with the conclusion that measured concentrations of coarse $Na^+$ and $Cl^-$ could mistakenly be ascribed to marine sources, rather than to locally originating fungal spore emissions. As discussed in Section 4.3.2, emission fluxes for $Cl^-$ are recorded throughout the campaign, and occurred during cooler, wetter periods at night. As noted by (Elbert et al., 2007), fungal spore emissions also predominantly occur under the same conditions. The

possibility that $Cl^-$ concentrations measured during this campaign are biogenically driven through the active discharge or rupturing of localized fungal spore emissions should therefore not be discounted.

### 4.4   Surface-atmosphere exchange of inorganic trace gases and aerosols

### 4.4.1   Dry deposition of HCl, HNO$_3$ and SO$_2$

As detailed in Section 3.3.1, HCl, $HNO_3$ and $SO_2$ were always deposited with no instances of emissions. The surface canopy

resistance ($R_c$) for these gases was calculated for the campaign using a rearranged form of Eq. (5). As expected on the basis of their high water solubility, $HNO_3$ and HCl deposited with a very small average canopy resistance of 1.42 and 2.92 $s\,m^{-1}$ respectively, which is not significantly different from zero given the typical uncertainty in the $R_b$ parameterisation used to infer this value. By contrast, the average canopy resistance for SO2 in this campaign was considerable, with a mean value of 86 $s\,m^{-1}$ throughout the entirety of the campaign, and a potentially more robust mean value of 28 $s\,m^{-1}$ for measurements

during daytime. Using his widely used dry deposition parameterisation, Wesely (1989) derives a typical $R_c$ value of 120 $s\,m^{-1}$ for $SO_2$ for deciduous forests with "lush vegetation" during "midsummer", evaluated at an incoming solar irradiance of 800 $W\,m^{-2}$. In the absence of tropical flux measurements, the appropriateness of the value for tropical forest has never been tested. While the observed average in this work is three times less than Wesely's parameterisation, the daytime average value from this campaign covers a wider set of meteorological conditions than used for the calculation of the modelled $SO_2$ $R_c$.





Zhang et al. (2003) elaborated upon Wesely's dry deposition parameterization through the development of a new formulation for the non-stomatal resistance component of the model. Modelled $v_d$ values for a variety of chemical species, including $SO_2$, were developed for different land use classifications (LUC), including broadleaf tropical forest. While Zhang et al. notes good agreement between modelled and observed $v_d$ values for LUCs such as short grasses and crops, the mean measured $v_d$ for $SO_2$ during this campaign deviates significantly from its corresponding modelled value for a tropical broadleaf LUC. This study

measured a mean $v_d$ of $10.4 \, \mathrm{mm \, s^{-1}}$ for $SO_2$, while Zhang et al. suggests values between 1.5 and $3.8 \, \mathrm{mm \, s^{-1}}$, with the limits for dry and wet canopies, respectively. As with the comparison with Wesely (1989), the appropriateness of modelled values have not been tested due to the lack of corresponding measurements. Similarly, the values for this campaign cover a wide range of meteorological conditions.

### 4.4.2 Bi-directional exchange of HONO and NH₃

Both HONO and $NH_3$ fluxes revealed periods of emission from the rainforest, with 26% of all HONO fluxes and 19% of $NH_3$ fluxes recorded as emissions. Due to the complexities of the chemical and physiological parameters controlling $NH_3$ emissions from the canopy surface to the atmosphere, discussion of the $NH_3$ fluxes measured in this study are considered in a separate paper (Ramsay et al., 2020), which investigates *inter alia* the influence of leaf wetness and modelled canopy compensation points upon $NH_3$ bi-directional exchange with reference to established models of $NH_3$ surface-atmosphere

exchange. It demonstrates that the observed $NH_3$ emissions are consistent with stomatal emission during the warmest part of the day and shows that measured leaf wetness is a more successful parameter in describing the cuticular deposition process than relative humidity and vapour pressure deficit. The present paper therefore focuses on discussion of the observed emissions of HONO at this site.

     The median diel fluxes of HONO in Figure 7 show emission in the early morning after dawn (from 07:00 to 09:00), with

deposition dominating throughout the rest of the day. Three possible explanations are considered here. The first considers the influence of soil emissions below the forest canopy. HONO emissions from soil have been observed in a number of studies (Sörgel et al., 2011, 2015; Twigg et al., 2011), with possible sources including the volatilization of HONO from soil nitrite (Su et al., 2011), the temperature-dependent activity of ammonia oxidizing bacteria (Oswald et al., 2013; Scharko et al., 2015), or the oxidation of hydroxylamine released from soil microorganisms (Ermel et al., 2018; Wu et al., 2019). During

night-time, radiative cooling above the forest causes stable stratification, generating a nocturnal boundary layer that prevents mixing between the air below and above the canopy (Foken, 2008; Tóta et al., 2008). Consequently, HONO emissions from the soil would accumulate below the canopy. At dawn, turbulent mixing starts to break up the nocturnal boundary layer, generating unstable conditions and a mixed layer. This creates a "venting" effect where the below-canopy accumulated HONO is transported upwards and appears as an early morning emission flux. Such venting episodes, representing negative storage

fluxes, are commonly observed for $CO_2$ over tall vegetation, and have been noted previously also in tower measurements above rainforests for $CO_2$ (Araújo et al., 2002), methane (Querino et al., 2011) and particles (Whitehead et al., 2010), with Querino et al. recording maximum median diel $CO_2$ and $CH_4$ fluxes between 06:00 and 10:00, similar to the period of maximum




median diel HONO emissions here. $CO_2$ flux measurements taken at the ATTO site concurrently with this study also showed a characteristic early morning flux, supporting the explanation of a "venting" effect for the HONO emissions.

However, morning HONO emissions have also been observed at short vegetation sites (Laufs et al., 2017; Di Marco et al., 2020; Ramsay et al., 2018), where storage effects are much smaller, and which therefore must have resulted from a different mechanism. This is that early morning HONO emissions are a consequence of the photolysis of $HNO_3$ (Zhou et al., 2011). Accumulation of $HNO_3$ on leaf surfaces during night-time result in a reservoir of $HNO_3$ within the canopy. At dawn, incoming solar radiation photolyses this reservoir, resulting in the formation of exited $NO_2$ radicals that—in the presence of photosen-

sitizing organics such as humic acid (George et al., 2005; Stemmler et al., 2007)—are reduced to HONO. The concurrent breakdown of the nocturnal boundary layer again results in an upward emission flux of HONO. However, while Zhou et al. (2011) recorded emissions of HONO from forests between the hours just after dawn until late afternoon, with maximum fluxes recorded around solar noon, in this study emissions occurred predominately during the hours immediately after dawn. While emissions were recorded at noon and during the afternoon on certain days, medial diel emissions were confined to 07:00 to

09:00 hours. Furthermore, Sörgel et al. (2015) has shown that this pathway would have a negligible effect on HONO formation based on the kinetic values for the pathway. Future work should measure the gradients of HONO above and below the canopy to determine whether HONO accumulation below canopy during stable night-time conditions is occurring, followed by venting during morning hours due to turbulent mixing.

    Finally, transient emission blips following sunrise have been observed for $NH_3$ during several studies, where they were

attributed to desorption of $NH_3$ that had been dissolved in dew and microscopic water layers overnight. As these water layers evaporate in the morning, concentrations increase to a point where they get driven into the gas phase. Studies (Di Marco et al., 2020; Rubio et al., 2008, 2002; He et al., 2006) have postulated that the same process occurs for HONO and contributes to the bi-directional exchange seen during some of the aforementioned observations. They show that timing is indeed consistent with the temporal dynamics of the emission at a UK grassland site. At ATTO, the temporal dynamics of the $NH_3$ flux were

different, with emission peaks occurring later in the day than for HONO and it was therefore concluded that desorption did not contribute to the $NH_3$ emission fluxes (Ramsay et al., 2020). It therefore remains unclear why desorption would have been more important for HONO than for $NH_3$.

    It is important to note that measurements of HONO by the GRAEGOR system are not artefact free. As detailed by Spindler et al. (2003), the presence of $SO_2$ and $NO_2$ on wet denuder walls can introduce a positive artefact that results in an overesti-

mate of HONO concentrations, which—if using a gradient system with two or more wet denuders set at different heights—can result in erroneous concentration gradient profiles. Correction algorithms exist for general application (Spindler et al., 2003) and specifically for the GRAEGOR (Ramsay et al., 2018) that allow the influence of the artefact to be quantified using concentrations of $SO_2$ and $NO_2$. However, for this campaign, no correction was necessary as the $SO_2$ concentration recorded during the campaign was 5 to 10 times lower than those relevant to artefact formation.





### 4.4.3 Deposition of water-soluble aerosols

The recorded deposition velocities of the aerosol species are consistent with the GRAEGOR/ACSM intercomparison: $NO_3^-$ and $Cl^-$ aerosols were predominantly contained in the coarse fraction, while $NH_4^+$ and $SO_4^{2-}$ were contained within the submicron aerosol. From a process-orientated approach (Davidson et al., 1982; Slinn, 1982; Slinn and Slinn, 1980), the deposition velocity of a particle is dependent upon its size. For particles $>0.1$ µm, deposition velocity (normalised against $u_*$) increases with increasing particle diameter. As outlined in Section 3.3.2, the close agreement between measured $SO_4^{2-}$ and $NH_4^+$ deposition velocities (and parametrised values for 0.1–0.2 µm size range aerosols) above tropical rainforest suggest that these aerosols were contained in the fine mode. These observed deposition velocities also agree well with modelled deposition velocities for $<1$ µm diameter particles above forest with similar mean roughness lengths and $u_*$ values as recorded at ATTO (Petroff et al., 2008). Conversely, the larger observed deposition velocities for $NO_3^-$ (5.8 mm s$^{-1}$) and $Cl^-$ (7.3 mm s$^{-1}$ exceed the parameterised values obtained using the formulation of Gallagher et al. (2002) and fit within the modelled values given by Petroff et al. (2008) for particles in the 2–10 µm range above surfaces with a similar roughness length.

As detailed in Section 3.3.2., occasional periods of apparent particle emissions from the rainforest were recorded throughout the campaign for all aerosol species measured. Deviations from near-exclusive deposition were rare (between 1%–3% of all measured fluxes), confined to one hour periods, and are unlikely to be due to measurement error. Similar to the emissions of HONO recorded during this campaign, upward particle fluxes may be caused by early morning turbulent mixing generating upward entrainment fluxes into the growing mixing layer. Whitehead et al. (2010) recorded a similar pattern of particle emissions at a tropical rainforest site in North Borneo, as did Ahlm et al. (2009) at a rainforest site in the Amazon basin located 120 km south-west of the ATTO site. However, both studies recorded a more predominant pattern of early morning emissions than here. Whitehead et al. (2010) recorded particle emissions for almost all mornings, while Ahlm et al. (2009) reported 40% of all particle fluxes as emissions. Both studies record later (08:00–09:00) emission periods. As both studies measured total particle number which was not chemically speciated, it is possible that the flux behaviour of the organic fraction of aerosol—which dominates the total aerosol mass fraction over tropical rainforest—is a more important driver for observed particle emissions than the aerosol species measured during this campaign.

### 4.5 Dry deposition budget of reactive nitrogen for the Amazon Rainforest based on dry season observations

The dry deposition of total reactive nitrogen to the ATTO site as derived from the GRAEGOR measurements ($\Sigma_{N_r} = NH_3 + NH_4^+ + HNO_3 + NO_3^- + HONO$) during this study relies on the assumption that values for $\Sigma_{N_r}$ in October are representative for the year overall. With this caveat, the annual dry deposition of $\Sigma_{N_r}$ for the ATTO site is estimated to be 1.7 kg N ha$^{-1}$ a$^{-1}$. The contribution of each reactive nitrogen species to this total is presented in Table 4.

Although dry deposition totals based on direct observation are rare for this biome, this estimate for dry $\Sigma_{N_r}$ should be considered as limited in scope due to the lack of a wet deposited $\Sigma_{N_r}$ value based on direct measurement. For example, (Trebs et al., 2006) previously reported that wet $\Sigma_{N_r}$ is the predominant contributor to total $\Sigma_{N_r}$ over the Amazon rainforest.





Furthermore, the present study's value of $\Sigma_{N_r}$ does not include water-soluble organic nitrogen (WSON), which can constitute up to 43% of total nitrogen in the aerosol phase during the dry season (Mace et al., 2003).

This study's $\Sigma_{N_r}$ dry deposition value of -1.7 kg N ha$^{-1}$ a$^{-1}$ based on dry season measurements is of the same order as
Trebs et al. (2006) equivalent estimate of -3.7 kg N ha$^{-1}$ a$^{-1}$ inferred from concentration measurements over a remote pasture site situated in the Amazon Basin. The stronger influence of agricultural activities and closer proximity of biomass burning at the pasture site in the Trebs et al. study may explains the slightly higher total $\Sigma_{N_r}$.

### 4.6    Comparisons of measured concentrations of trace gases and associated aerosols with previous studies

Whilst this was a one-month study limited to the dry season, during which local, regional and global biomass burning con-
tributed to observed concentrations, it provides some insight into the atmospheric composition of an ecosystem for which there are few measurements overall. Placing these measurements in context with similar regional and local studies above tropical rainforest sites provides an impression of the spatial and temporal representativeness of this study.

For aerosols, measurements of PM$_{10}$ concentrations (both cations and anions) taken by high-volume air samplers between 2008 and 2016 over the Cuieiras ZF2 natural reserve approximately 130 km west of the ATTO site have recently become
available (Custodio et al., 2019), allowing a local comparison in measured aerosol concentrations between the GRAEGOR and filter sampling. For Cl$^-$ and NO$_3^-$, the average measurements taken by the GRAEGOR are between 2.5 and 4 times greater than the average from 10 samples collected by the high volume air samplers during the dry seasons in the period 2008 to 2016. Conversely, the average dry season SO$_4^{2-}$ concentrations recorded by the GRAEGOR is 0.3 times that recorded by the high volume samplers. NH$_4^+$ concentrations recorded by both measurement techniques are approximately equivalent.

Measurements of aerosol composition taken during the Amazon Boundary Layer Experiment (ABLE-2A) (Talbot et al., 1988) provide mean concentration values for the same species measured during this study. Talbot et al. measures a mean atmospheric concentration in the mixed layer for NH$_4^+$ as 12 nmol$^{-3}$ or 0.22 µg m$^{-3}$; and for SO$_4^{2-}$ as 5.2 nmol$^{-3}$ or 0.5 µg m$^{-3}$. These values are higher than those measured in this study (mean concentration of NH$_4^+$ = 0.16 µg m$^{-3}$, and SO$_4^{2-}$ = 0.25 µg m$^{-3}$). In comparison, the mean concentrations measured during ABLE-2A of NO$_3^+$ (4.4 nmol$^{-3}$ or 0.22 µg m$^{-3}$) and
Cl$^-$ (1.2 nmol$^{-3}$ or 0.04 µg m$^{-3}$) are lower than those measured during this study.

Discrepancies in the measurements of these aerosol species between wet-chemistry instruments and high-volume air-sampler systems have previously been noted by Trebs et al. (2008), who found a similar order of magnitude difference in SO$_4^{2-}$ measurements between a WRD-SJAC system and a high volume air sampler in tropical conditions. They also reported that high-volume air samplers measured lower concentrations of Cl$^-$ and NO$_3^-$ compared to wet chemistry instruments, although
this pattern was only observed during periods of low concentrations of Cl$^-$ and NO$_3^-$. Loss of Cl$^-$ and NO$_3^-$ from high volume filters has been reported frequently, and this issue in fact led to the development of the SJAC sampling system, which does not suffer from this artefact (Slanina et al., 2001). Trebs et al. (2008) attributed higher SO$_4^{2-}$ high-volume air-sampler concentrations to the decomposition of organosulfates on filters during storage, as well as to environmental conditions such as high relative humidity that may have introduced both positive and negative artefacts on the filter substrate.



The most comprehensive previous report of $NH_3$, $SO_2$ and $HNO_3$ concentrations over remote tropical rainforests is by Adon et al. (2010), who presented long-term measurements over Cameroonian rainforest using passive denuder tubes. For the dry season, Adon et al. reported a similar concentration of $SO_2$ and $HNO_3$, but reported a significantly higher concentration of $NH_3$ (a dry season average of $2.9\ \mu g\,m^{-3}$ compared to $0.28\ \mu g\,m^{-3}$ reported in this study). Adon et al. postulated that the $NH_3$ concentrations recorded over their rainforest site were driven by biomass burning, similar to the conclusion drawn in this

study. It is possible that the intensity, proliferation and proximity of biomass burning at the Cameroonian site may therefore be heightened in comparison to the ATTO site, resulting in greater measurements of $NH_3$ concentrations.

Trebs et al. (2004), using a wet annular rotating denuder with steam jet aerosol collector system – effectively a single-height GRAEGOR instrument – measured the same suite of inorganic trace gases and associated aerosols as this study, but at a pasture site located in the southern Amazon Basin. Measurements in the dry season had similar mean and median concentrations of

$NH_3$ and $HNO_3$ as this study, but higher concentrations of HCl and HONO, and with $SO_2$ having the lowest concentration of the inorganic trace gases measured. As a fractional contribution to acid loading, this suggests that HCl is even more dominant than at the ATTO site, which is expected for an active pasture site with local biomass burning compared with the ATTO pristine rainforest site.

## 5   Conclusions

This study employed a two-point, wet-chemistry instrument (GRAEGOR) to measure online, hourly-resolved concentrations and fluxes of the inorganic trace gases $NH_3$, HCl, HONO, $HNO_3$ and $SO_2$ and their associated water-soluble aerosol counterparts $NH_4^+$, $Cl^-$, $NO_2^-$, $NO_3^-$ and $SO_4^{2-}$ for a 1-month period over the Amazon rainforest. While measurements of $NO_2^-$ aerosol concentrations were below the detection limit, this study presents for the first time the concentrations, fluxes and deposition velocities for several species during the Amazon dry season. This study has also confirmed the applicability of the Chor

et al. (2017) flux enhancement factor ($\gamma_F$) for correcting fluxes measured using the aerodynamic gradient method within the roughness sublayer above tropical rainforest. Some of the key findings are summarised below:

1. Influence of local, regional and potentially global transport of pollutants. Elevated concentrations of $SO_2$ and $NH_3$, together with $BC_e$ and $c_{CO}$ proxies for anthropogenic emissions, were noted at several points during the campaign. Back trajectory analysis for particularly polluted conditions showed that air masses arriving at the ATTO site during

795       this period travelled over large urban areas to the south and south-east of the site, as well as over areas with fires. For some air masses during the polluted periods of the campaign, air-mass trajectories were recorded which originated along the coast or interior of south west Africa. This area is a location of biomass burning during the August-October period. Long-range transport episodes, driven by African biomass burning, could therefore contribute to an overall background of increased pollution during the Amazon dry season.

2. Bi-directional exchange of inorganic trace gases and aerosols. While the gases HCl, $HNO_3$ and $SO_2$ were uniformly deposited to the rainforest canopy, 26% of all HONO fluxes and 19% of $NH_3$ fluxes were recorded as emissions. For





HONO and the aerosol species, the occurrence of venting—whereby the accumulation of a gas or aerosol species below or on the canopy is swiftly entrained into the mixed layer through early morning turbulence—is suggested as an explanation for the instances of emission.

3. Influence of coarse aerosol on total aerosol fraction above Amazon rainforest. This study presents the first online measurements of chemically-speciated aerosol concentration in total suspended particulates and, by comparison with the ACSM, in the coarse fraction. The contribution of $Cl^-$ and $NO_3^-$ to the total aerosol mass is substantially higher than in the sub-micron fraction and concentrations of both components are significantly larger than had previously been estimated on the basis of ACSM and AMS measurements. The deposition velocities of $Cl^-$ and $NO_3^-$ aerosol were

consistent with their being predominantly in the coarse size fraction. The presence of coarse aerosol at the ATTO site could be derived from a combination of sources, including biomass burning point sources within the region, from seasalt advected to the site by intrusions of marine air, and from biogenic crustal material such as fungal spores.

An estimate of total reactive nitrogen dry deposition ($\Sigma_{N_r} = NH_3 + NH_4^+ + HNO_3 + NO_3^- + HONO$) for the Amazon rainforest has also been presented, on the basis that these dry season measurements are representative for the total year. The

estimated annual value for $\Sigma_{N_r}$ based on measurements was -1.7 kg N ha$^{-1}$ a$^{-1}$, a net deposition of reactive nitrogen to the rainforest with the largest contributor being $NH_3$, contributing 0.74 kg N ha$^{-1}$ a$^{-1}$ to the overall total. This value presents the first estimate for reactive nitrogen dry deposition to rainforests based on in-situ measurements of reactive nitrogen species. Our results show that dry deposition is of similar magnitude as earlier estimates of wet deposition. For example, Lesack and Melack (1996) estimated a wet deposition value of +2.4 kg N ha$^{-1}$ a$^{-1}$ for total nitrogen, which includes particulate nitrogen

and dissolved organic nitrogen, while Andreae et al. (1990b) estimated a wet deposition flux of 2.1 kg N ha$^{-1}$ a$^{-1}$ in the form of ammonium and nitrate.

The measurements presented here confirm the importance of measuring chemically-speciated inorganic trace gases and associated aerosols above rainforest as, by doing so, important atmosphere-exchange processes (venting from the forest floor, increased deposition during pollution episodes) and knowledge of aerosol speciation (the importance of the coarse mode on

total aerosol mass) become apparent. With the implementation of the ATTO 325-m tower, the potential now exists for further long-term measurements of inorganic trace gases and aerosols using GRAEGOR or commercial GRAEGOR derivatives (such as the Monitor for Aerosols and Gases in Ambient Air, MARGA, Metrohm Applikon). Replicating this study in the wet season, or by including measurements of the concentrations and fluxes of water soluble organic nitrogen through modifications to the GRAEGOR, are potential avenues for future investigation.

*Author contributions.* EN, CDFM, MRH, MS, PA and MA devised the study and secured the funding. GRAEGOR measurements were taken by RR and CDFM. GRAEGOR data was processed by RR with input from CDFM, EN, MRH, and MS. ToF-ACSM measurements were taken by SC. $BC_e$ and $c_{CO}$ measurements were taken by CP and JL. AA and MS provided ancillary measurement data, including micrometeorological data. RR interpreted the data with contributions from EN, CDFM, MRH, MS and MA. RR led the manuscript with contributions from all the authors.





*Competing interests.* The authors declare that they have no conflict of interest.

*Acknowledgements.* This work was enabled through a studentship funded jointly by the University of Edinburgh School of Chemistry and the Max Planck Institute for Chemistry. CDM, EN and the GRAEGOR instrument were supported by the UK Natural Environment Research Council award number NE/R016429/1 as part of the UK-SCAPE programme delivering National Capability. We thank the Instituto Nacional de Pesquisas da Amazonia (INPA) and the Max-Planck Society for continuous support. We acknowledge the support by the
German Federal Ministry of Education and Research (BMBF contract 01LB1001A and 01LK1602B) and the Brazilian Ministério da Ciência, Tecnologia e Inovação(MCTI/FINEP contract 01.11.01248.00) as well as the Amazon State University (UEA), FAPEAM, LBA/INPA and SDS/CEUC/RDS-Uatumã. We acknowledge funding from FAPESP (Fundação de Amparo à Pesquisa do Estado de São Paulo) trough grant 2017/17047-0. We acknowledge the use of data and imagery from LANCE FIRMS operated by NASA's Earth Science Data and Information System (ESDIS) with funding provided by NASA Headquarters. The authors are grateful for the support of the Amazon Tall
Tower Observatory staff and visiting researchers. In particular, the authors would like to thank Mr. Reiner Ditz, Mr. Andrew Crozier, Dr. Stefan Wolff, Mr. Pedro Assis and Ms. Isabella Hrabe de Angelis for their support throughout the campaign.



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



**Table 1.** Mean ($\mu_A$), median ($\mu_M$), arithmetic standard deviation ($\sigma_A$), maximum, minimum and number of measurements for water soluble aerosol and inorganic trace gas concentration measurements taken at 60 m on the 80-m tower, with associated limits of detection (LOD) values for each species based on 30-minute values.

| (60 m) | $\mu_A$ $\mu g\,m^{-3}$ | $\mu_M$ $\mu g\,m^{-3}$ | $\sigma_A$ $\mu g\,m^{-3}$ | Max $\mu g\,m^{-3}$ | Min $\mu g\,m^{-3}$ | No. of Measurements | LOD $\mu g\,m^{-3}$ |
|---|---|---|---|---|---|---|---|
| $NH_4^+$ | 0.30 | 0.30 | 0.16 | 0.73 | 0.01 | 508 | 0.19 |
| $Cl^-$ | 0.23 | 0.14 | 0.22 | 1.3 | 0.01 | 516 | 0.01 |
| $NO_2^-$ | 0.01 | 0.01 | 0.01 | 0.09 | 0.00 | 577 | 0.02 |
| $NO_3^-$ | 0.47 | 0.41 | 0.33 | 2.1 | 0.05 | 489 | 0.16 |
| $SO_4^{2-}$ | 0.51 | 0.49 | 0.25 | 1.1 | 0.07 | 528 | 0.1 |
| $NH_3$ | 0.28 | 0.25 | 0.18 | 1.9 | 0.01 | 558 | 0.17 |
| HCl | 0.13 | 0.11 | 0.09 | 0.47 | 0.03 | 526 | 0.07 |
| HONO | 0.07 | 0.06 | 0.04 | 0.38 | 0.01 | 599 | 0.03 |
| $HNO_3$ | 0.25 | 0.23 | 0.14 | 1.0 | 0.03 | 579 | 0.12 |
| $SO_2$ | 0.23 | 0.21 | 0.11 | 0.84 | 0.01 | 549 | 0.10 |





**Table 2.** Mean ($\mu_A$), median ($\mu_M$), maximum and minimum values post- roughness sub layer correction for fluxes, deposition velocities ($V_d$), theoretical maximum deposition velocities ($V_{max}$) and canopy resistances ($R_c$) for the inorganic trace gases measured during Amazon Tall Tower Observatory campaign. The number of fluxes calculated is quoted as number of measurements, and the median error in flux measurements as a percentage of flux values for each individual trace gas species ($\sigma_F$) is included as part of the statistical summary for fluxes.

|  |  | NH$_3$ | HCl | HONO | HNO$_3$ | SO$_2$ |
|---|---|---|---|---|---|---|
| Flux (ng m$^{-2}$ s$^{-1}$) | $\mu_A$ | −2.8 | −2.3 | −0.34 | −3.6 | −2.4 |
|  | $\mu_M$ | −1.8 | −1.4 | −0.23 | −2.3 | −1.2 |
|  | Max | 9.5 | 0.67 | 4.0 | 2.4 | 1.2 |
|  | Min | −30 | −17 | −7.1 | −25 | −33 |
|  | No. of measurements | 434 | 400 | 422 | 405 | 405 |
|  | $\sigma_F$ (%) | 33 | 56 | 54 | 45 | 63 |
| $V_d$ (mm s$^{-1}$) | $\mu_A$ | 10.5 | 15.2 | 4.5 | 12.4 | 10.4 |
|  | $\mu_M$ | 8.3 | 14.3 | 4.1 | 11.9 | 7.1 |
|  | Max | 80 | 79 | 64 | 63 | 74 |
|  | Min | -36 | -9.8 | -141 | -22 | -3.4 |
| $V_{max}$ (mm s$^{-1}$) | $\mu_A$ | 19.3 | 15.3 | 12.6 | 12.3 | 12.9 |
|  | $\mu_M$ | 18.1 | 14.5 | 12.1 | 11.9 | 12.4 |
|  | Max | 50 | 39 | 31 | 31 | 32 |
|  | Min | 0.75 | 0.60 | 0.49 | 0.49 | 0.52 |
| $R_c$ (s m$^{-1}$) | $\mu_A$ | 52 | 2.9 | 165 | 1.4 | 86 |
|  | $\mu_M$ | 64 | 1.6 | 165 | 1.8 | 33 |





**Table 3.** Mean ($\mu_A$), median ($\mu_M$), maximum and minimum values post- roughness sub layer correction for fluxes and deposition velocities ($V_d$) for the water soluble aerosols measured during the Amazon Tall Observatory campaign. The number of fluxes calculated is quoted as number of measurements, and the median error in flux measurements as a percentage of flux values for each individual aerosol species ($\sigma_F$) is included as part of the statistical summary for fluxes.

| | | $NH_4^+$ | $Cl^-$ | $NO_3^-$ | $SO_4^{2-}$ |
|---|---|---|---|---|---|
| Flux ($ng\,m^{-2}\,s^{-1}$) | $\mu_A$ | -1.7 | -2.3 | -4.4 | -3.5 |
| | $\mu_M$ | -1.2 | -1.2 | -2.7 | -2.8 |
| | Max | 0.70 | 3.6 | 2.9 | 4.3 |
| | Min | -11 | -23 | -24 | -22 |
| | No. of measurements | 427 | 371 | 342 | 360 |
| | $\sigma_F$ (%) | 56 | 43 | 44 | 41 |
| $V_d$ ($mm\,s^{-1}$) | $\mu_A$ | 2.9 | 7.8 | 7.0 | 3.7 |
| | $\mu_M$ | 2.6 | 7.3 | 5.8 | 2.8 |
| | Max | 25 | 54 | 49 | 33 |
| | Min | -2.6 | -12 | -8.1 | -7.6 |





**Table 4.** Contribution of reactive nitrogen species to total ($\Sigma$ ($NH_3$ + $NH_4^+$ + $HNO_3$ + $NO_3^-$ + HONO)) reactive nitrogen dry deposition budget for ATTO in $\mathrm{kg\,N\,ha^{-1}\,a^{-1}}$, inferred from fluxes measured during campaign.

| Reactive Nitrogen Species | $\mathrm{kg\,N\,ha^{-1}\,a^{-1}}$ |
|---|---|
| $NH_3$ | -0.74 |
| HONO | -0.03 |
| $HNO_3$ | -0.25 |
| $NH_4^+$ | -0.41 |
| $NO_3^-$ | -0.31 |
| $\Sigma_{N_r}$ = $NH_3$ + $NH_4^+$ + $HNO_3$ + $NO_3^-$ + HONO | -1.7 |





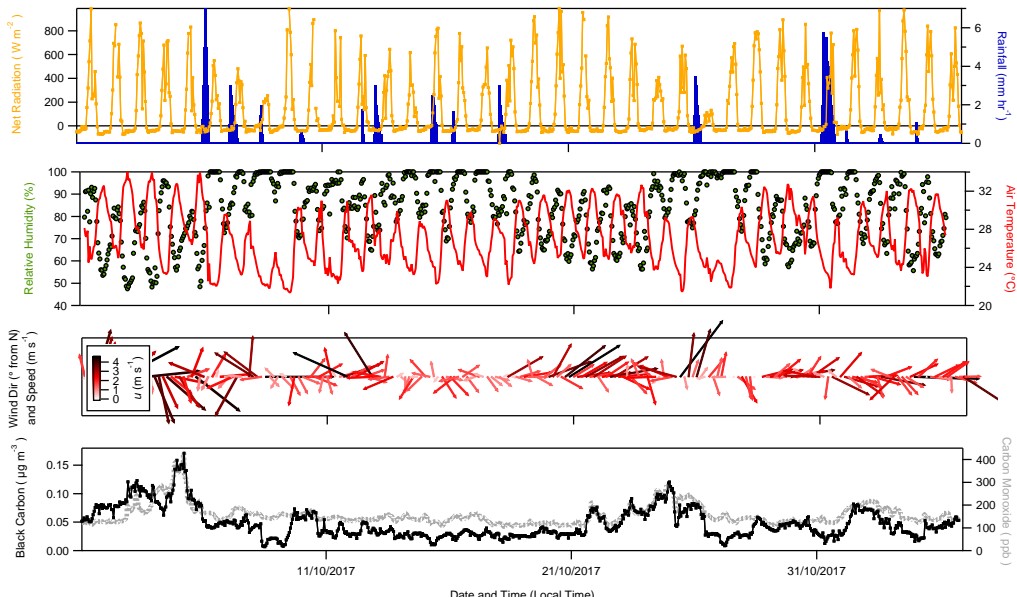

**Figure 1.** Meteorological and supplementary measurements taken during the campaign. From top, net radiation, hourly rainfall, relative humidity, air temperature, wind speed and wind direction (barbs scaled to wind speed, and orientated from 0° North), and concentrations of black carbon and carbon monoxide.





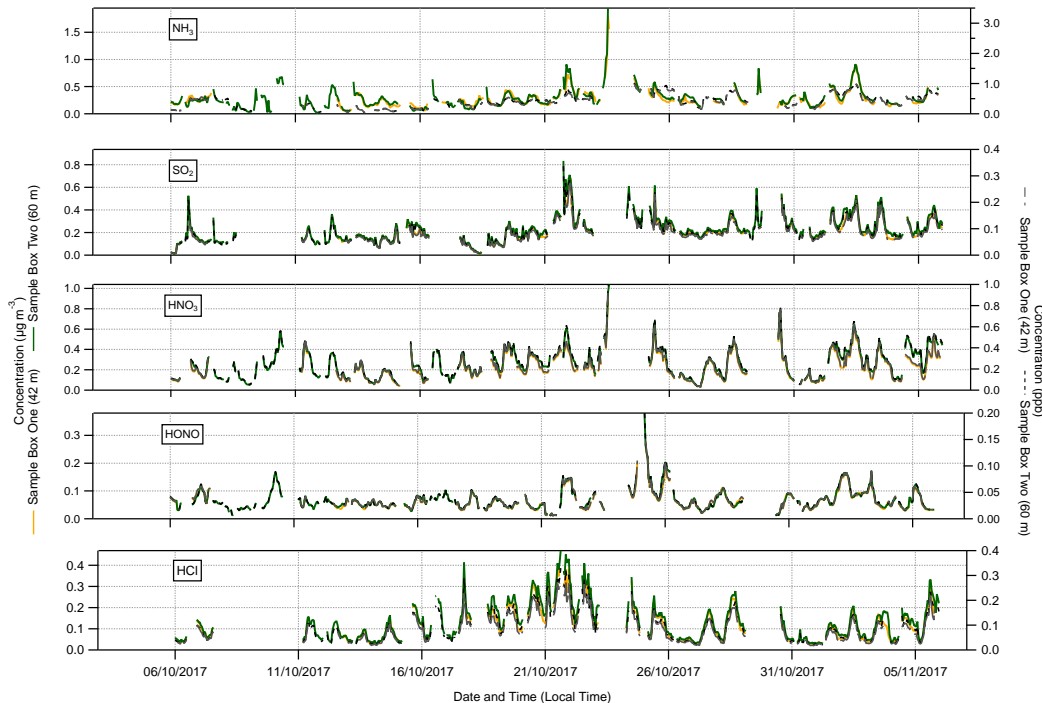

**Figure 2.** Time series of hourly concentrations (primary left axis, mass concentrations; secondary right axis, molar mixing ratios) of inorganic trace gas species measured by the GRAEGOR at 42 m (yellow) and 60 m (green) on the 80-m tower at the Amazon Tall Tower Observatory site.

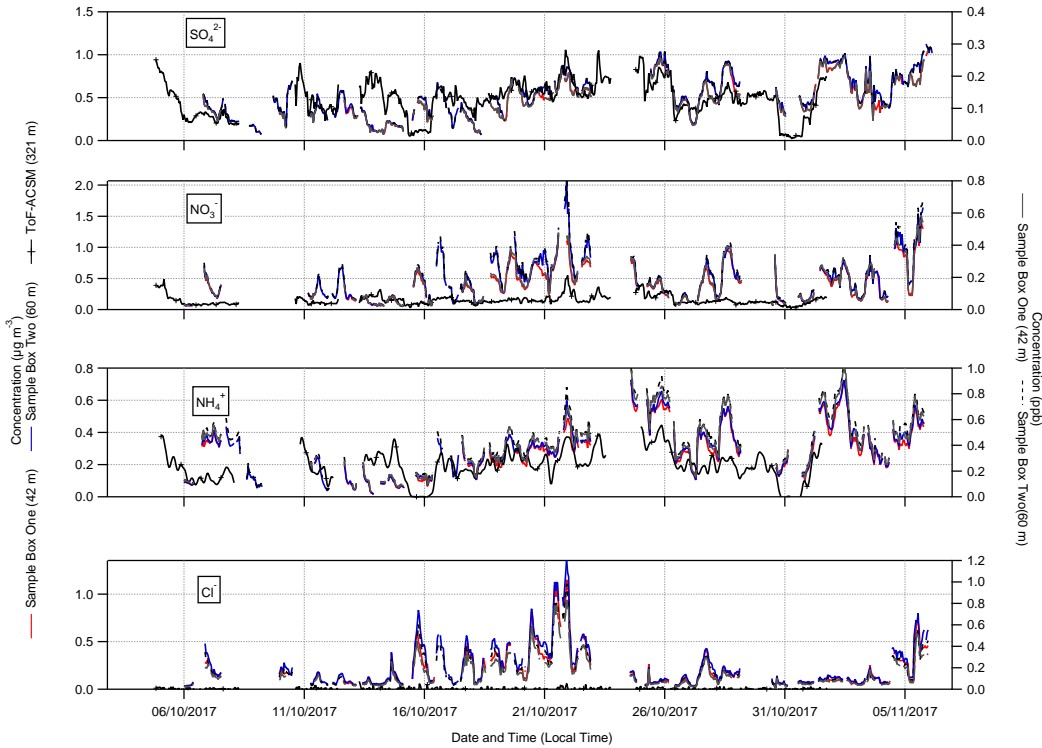

**Figure 3.** Time series of hourly concentrations (primary left axis, mass concentrations; secondary right axis, molar mixing ratios) of water-soluble aerosol species measured by the GRAEGOR at 42 m (red) and 60 m (blue) on the 80-m tower, and ToF-ACSM at 321 m (black) at the Amazon Tall Tower, at the Amazon Tall Tower Observatory site.



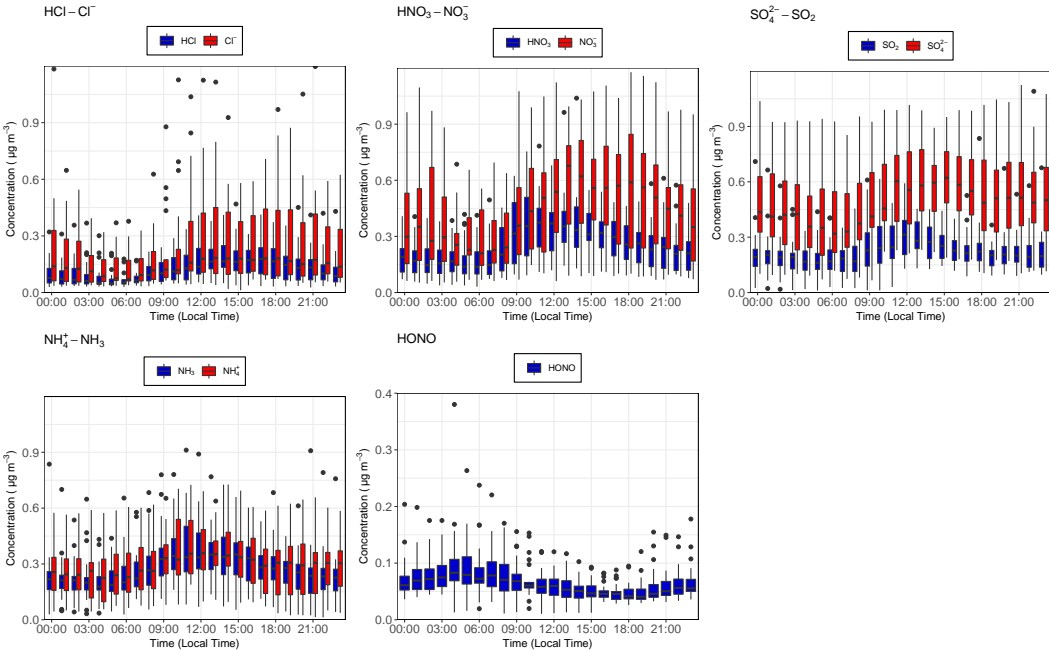

**Figure 4.** Median hourly diel concentrations for the inorganic trace gases $NH_3$, $SO_2$, HONO, $HNO_3$ and HCl in blue, and their paired associated aerosol counterparts $NH_4^+$, $SO_4^{2-}$, $NO_3^-$ and $Cl^-$ in red at the 60 m sampling height measured during the campaign.





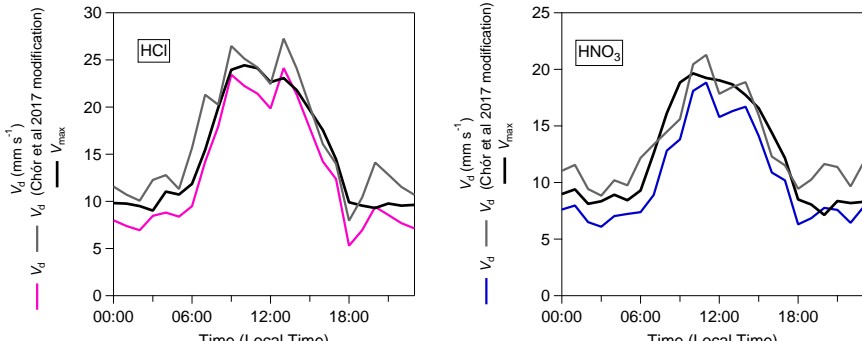

**Figure 5.** Inorganic trace gas deposition velocities ($V_d$) pre- and post- correction with $\gamma_F$ (Chor et al., 2017) and calculated theoretical maximum deposition velocities ($V_{max}$) for HCl and HNO$_3$.





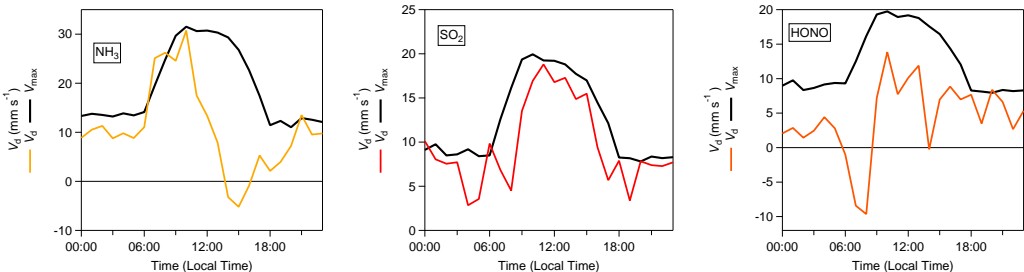

**Figure 6.** Inorganic trace gas deposition velocities ($V_d$) pre- and post- correction with $\gamma_F$ (Chor et al., 2017) and calculated theoretical maximum deposition velocities ($V_{max}$) for $NH_3$, $SO_2$ and HONO.



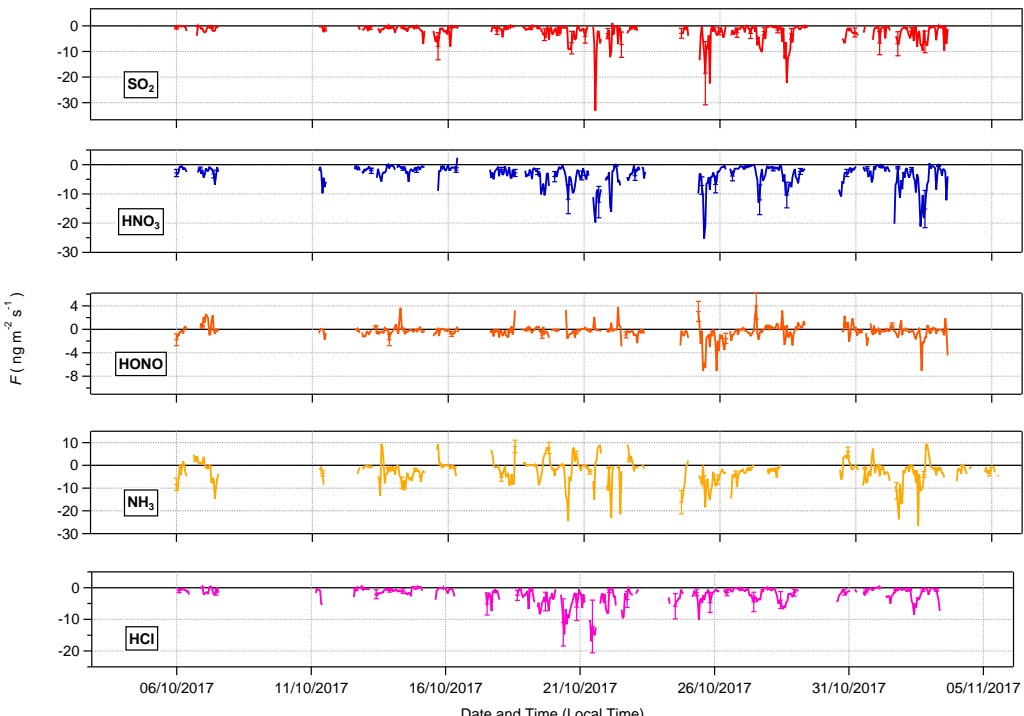

**Figure 7.** Time series of filtered fluxes for the inorganic trace gas species measured during the campaign.





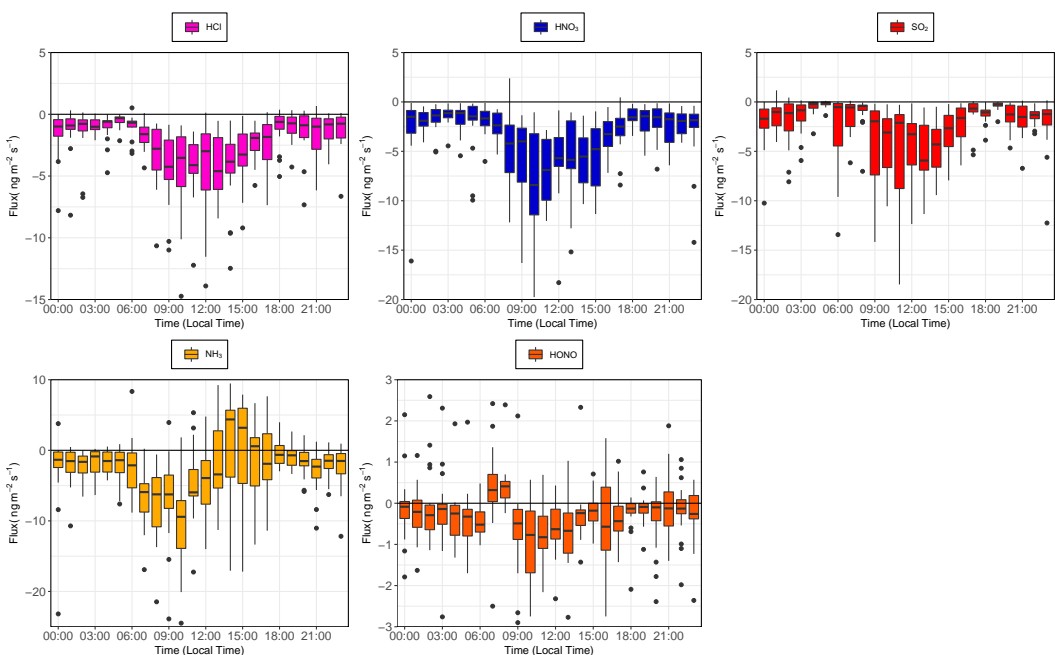

**Figure 8.** Calculated median diel fluxes of inorganic trace gas species measured during the campaign. From top left (clockwise) – HCl, $HNO_3$, $SO_2$, HONO and $NH_3$.





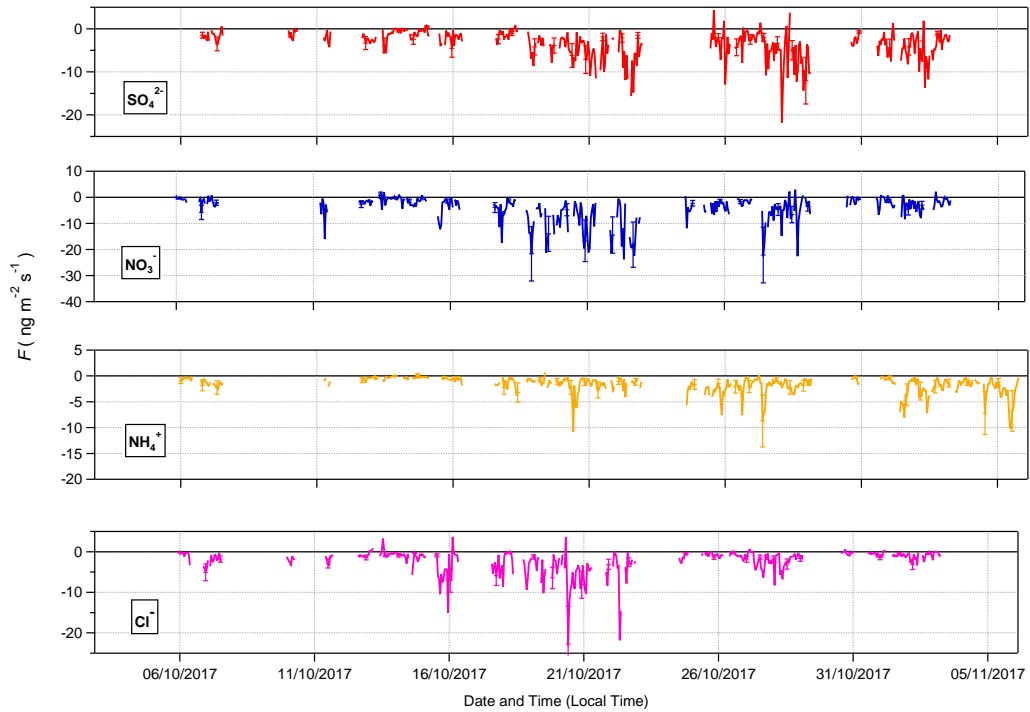

**Figure 9.** Time series of filtered fluxes for the aerosol counterpart species measured during the campaign.



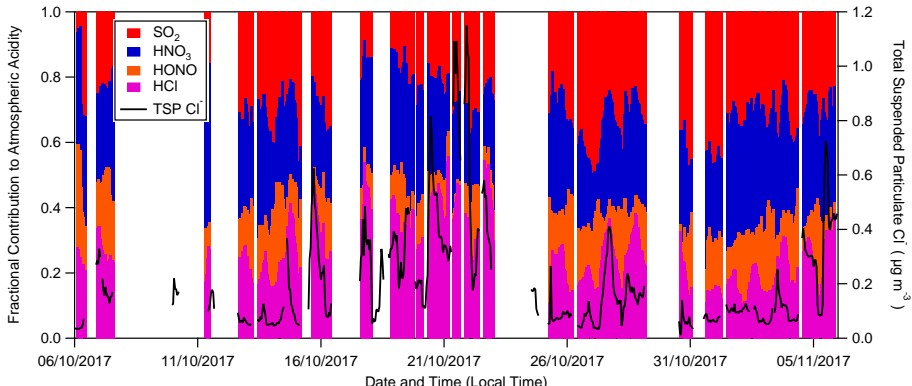

**Figure 10.** Fractional contribution to total measured inorganic acidity from $SO_2$, $HNO_3$, HONO and HCl as measured by the GRAEGOR at 60 m (hourly resolution). The concentration of total suspended particulate $Cl^-$ is included as an indicator of periods where sea salt or chloride containing particulate was present at the ATTO site.



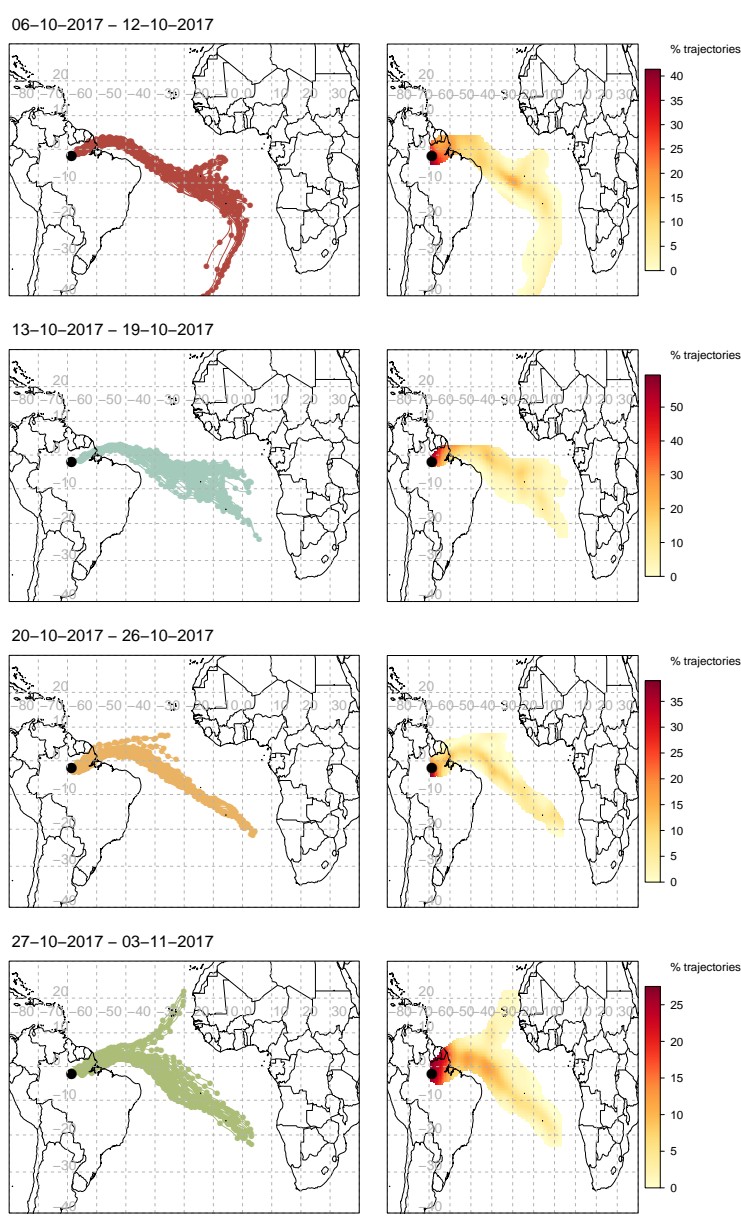

**Figure 11.** Air-mass back-trajectories arriving at the 80-m walk up tower on each day every three hours from 00:00 local time over the period from 6 October 2017 to 3 November 2017, grouped by week, and coupled with associated frequency trajectory plots. The duration of each trajectory is 10 days, marks indicate 12-hour intervals. Modelled using NOAA HYSPLIT 4 using GDAS1 meteorology.





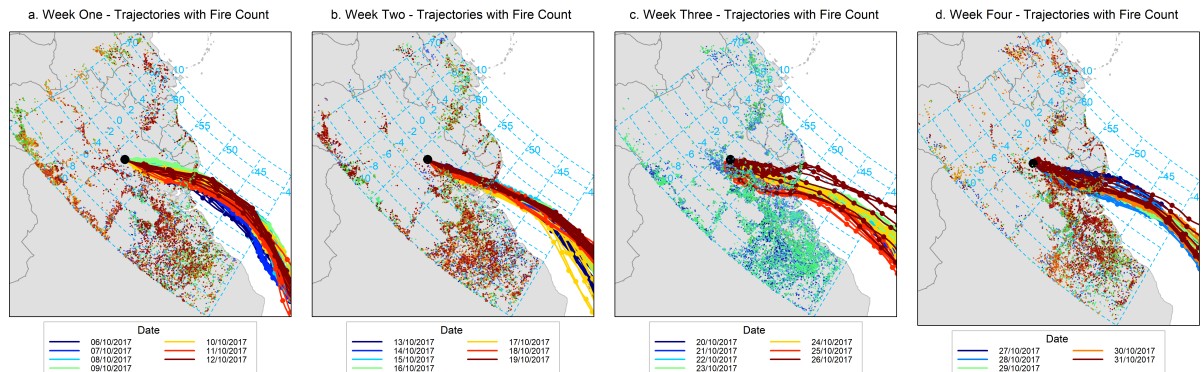

**Figure 12.** Air-mass back-trajectories arriving at the 80-m walk up tower on each day every three hours from 00:00 local time over the period from 6 October 2017 to 31 October 2017, grouped by week, and further subdivided by day, for the regional area surrounding the ATTO site. Fire count data is included as an overlay to each weekly plot, with fire count coloured according to the date on which the fire was recorded by satellite imagery.





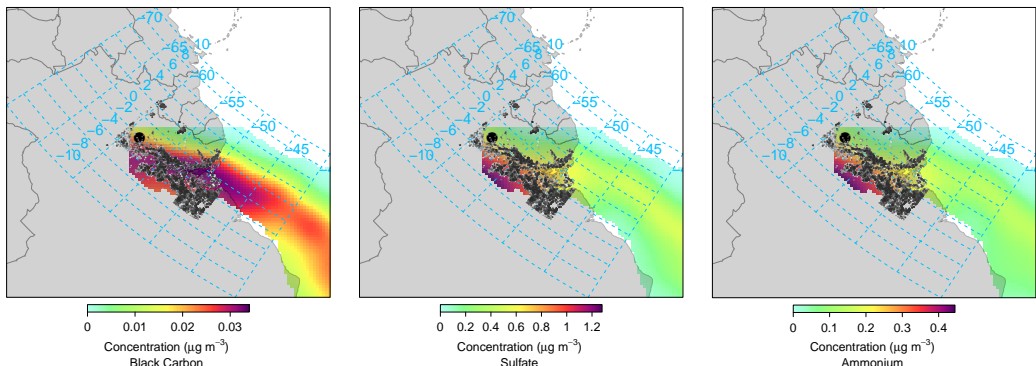

**Figure 13.** Concentration weighted trajectory analysis for (from left) $BC_e$, $SO_4^{2-}$ and $NH_4^+$, with fire data overlaid. Fire data is coloured (scale, from light grey to black) by fire intensity, a measure of the fire radiative power of the individual fire.



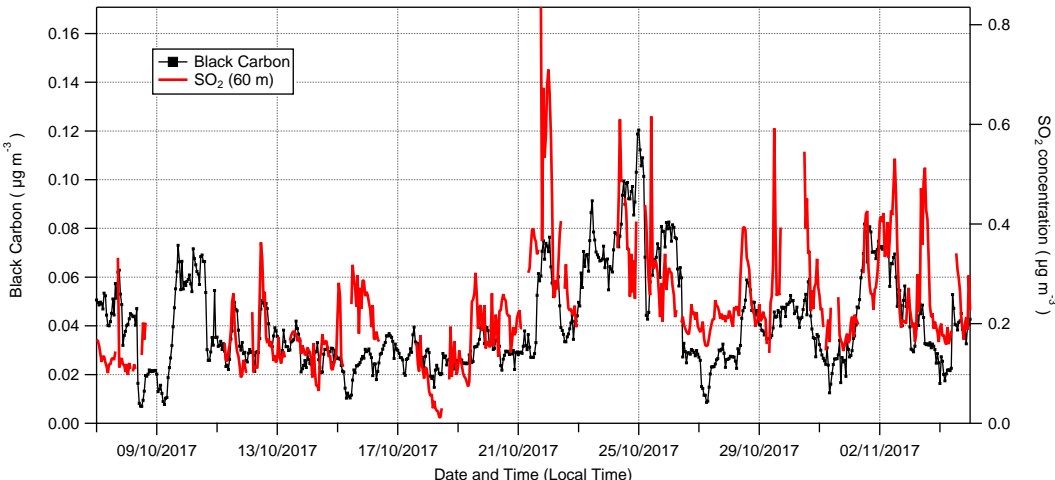

**Figure 14.** Time series of hourly $SO_2$ and $BC_e$ concentrations, highlighting the close correlation between $SO_2$ and $BC_e$ measurements throughout the campaign.



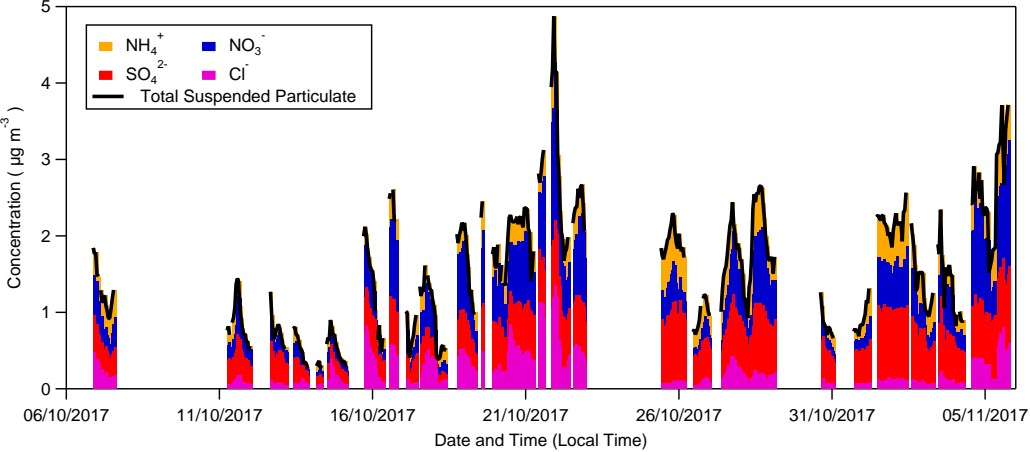

**Figure 15.** Summed mass and speciation of suspended particulate recorded by the GRAEGOR at 80-m throughout period of campaign.