# Peer review of "Concentrations and biosphere-atmosphere fluxes of inorganic trace gases and associated ionic aerosol counterparts over the Amazon rainforest"

_Atmospheric Chemistry and Physics, 2020_

## Referee Comment (RC1) · Anonymous Referee #1 · 14 Sep 2020

This study present a systematic high-resolution measurement of concentrations, fluxes and deposition velocities of the inorganic trace gases and their water-soluble aerosol counterparts during the dry season of the Amazon rainforest. This study provides a highly useful dataset for atmospheric chemistry studies over the tropical rainforest, especially considering the lack of systematic measurements in such environments. The paper is generally well written. I suggest that it be accepted for publications after the authors make the following minor revisions.

1) Line 2-4: The wet season can also be affected by anthropogenic pollution, such as

that from Manaus.

2) Line 354-360: What are the possible reasons for the large difference between GRAEGOR and ToF-ACSM measurements for nitrate and chloride?

3) Line 779-780: Trebs et al. (2004) shows an average NH3 concentration of about 2 ppb in the dry season, which is about 4 times larger than the measurement of this study (about 0.5 ppb).

4) Many texts in the figures are quite small. You need to make sure that all texts are legible without zooming in.

5) Figs. 2 and 3: You should also describe the black dashed lines and grey dashed lines in the figure captions.

6) Figs. 4 and 8: Please describe the meanings of each component of the boxplots.

7) Fig. 6: This figure does not show pre-correction values as suggested by the caption.

8) Fig. 14: This figure is very similar to Fig. 1d except that an SO2 line is added. I think it can be simply combined into Fig. 1. Also, I am wondering why the period before October 7 is not shown in Fig. 14.

9) Fig. 15: I think "total suspended particulate" is not an appropriate term here because only a few inorganic components are included. Sometimes you used "water-soluble total suspended particulate" in the text. This is better but still not accurate since some organic components are also water-soluble.

ACPD

---

## Referee Comment (RC2) · Anonymous Referee #2 · 14 Sep 2020

**Review of manuscript titled "Concentrations and biosphere-atmosphere fluxes of inorganic trace gases and associated ionic aerosol counterparts over the Amazon rainforest" by Ramsay et al. submitted to EGU's *Atmospheric Chemistry and Physics***

This work provides, high temporal resolution (hourly) surface-atmosphere flux and deposition products such as concentration, flux and deposition velocities of inorganic trace gases and their aerosol counterparts, not routinely or simultaneously measured previously over a tropical rainforest. Provided, deposition velocities and fluxes used for dry deposition parametrization in modeling studies for tropical forest have been interpolated so far from temperate forest observations. This work is pertinent for understanding boundary-layer chemistry and improving dry deposition and flux estimates of inorganic species discussed ($NH_3$, HONO, $HNO_3$, $SO_2$ and their corresponding aerosol counterparts) in modeling studies, specifically for tropical forests. However, the scope of this study is limited to the 2017 dry season (October-November) in Amazon rain forest as stated in the manuscript. Authors point towards the need of future work expanding to wet season to understand annual pattern of surface-atmosphere exchanges in tropical forests.

The manuscript is well written with findings presented well through descriptive statistics and visualizations. The findings are critical for a wider regional- and global- scale modeling community interested not just in surface-atmosphere interactions but in specific aspects such as, Nitrogen deposition critical to biosphere. I will encourage this manuscript for publication, once authors address the following edits/comments:

1) As mentioned in the manuscript: "Based on the height of the tallest trees, the canopy height ($h_c$) is 37.5 m (Chor et al., 2017)." While, for this study two heights used for gradient measurements on the 80-m walk up tower were: z1= 42 m and z2= 60 m, *both are above-canopy*. Also Fig. 2 exhibits marginal difference in hourly concentrations measured at these two sampling heights above canopy. Can authors elaborate more on any limitations on doing measurements at a sampling height $< h_c$ (i.e. < 37.5 m)? Concentrations between above-canopy and below-canopy sampling heights would have shown more substantial gradient and possibly given better insights on canopy reductions and boundary layer chemistry of different species in a tropical rain forest? Is that something that can be focused in a future study (as discussed for HONO briefly in Lines 680-690)?

2) Please consider shifting some figures to supplement or split up some busy figures. For instance, if you want to keep the molar mixing ratios shown on secondary axes of Figs. 2 and 3, might be better to show them separately in supplement for clarity. Also increase text size of labels in figures wherever possible.

3) Line 340: correct 'HNO)3' to $HNO_3$.

4) Lines 348-360 and Figure 3: Authors mention there is a significant difference between GRAEGOR and ToF-ACSM measurements for both $NO_3^-$ and $Cl^-$. Maybe consider adding any linear regression analysis as done for $SO_4^{2-}$ and $NH_4^+$ to reach that conclusion (to ensure it is not a scale issue- $NO_3^-$ and $Cl^-$ ToF-ACSM measurements being order of magnitudes lower than GRAEGOR) ?

   More discussion on how different sampling heights of GRAEGOR and ToF-ACSM measurements (60 m and 321 m respectively)  matter, might help? Consider merging the argument made in Lines 427-429 to explain the GRAEGOR vs TOF-ACMS difference for : "…$V_{ds}$ with increasing particle size, the larger median $V_{ds}$ values for $Cl^-$ and $NO_3^-$ are consistent with the GRAEGOR vs ACSM comparison which suggests that these aerosol counterparts were present in the super-micron (>$PM_1$) fraction."

5) Lines 362-365: "Although the diel cycle of HONO exhibited a maximum during night and a minimum during the day (0.02 µg m$^{-3}$ at 14:00), it remained above the detection limit even during daylight hours (Figure 4), which, given the high photolysis rate of HONO during daytime, implies the presence of a daytime source." Can this day-time source of HONO point to biogenic soil HONO emissions? How would they compare to anthropogenic sources in 'polluted' conditions?

6) Figure 5: Please clarify in the caption that pre- and post- correction deposition velocity trends are denoted by colored and grey lines respectively or use same convention in Fig. 5 labels and caption.

7) Line 390: "……although the results would be sensitive to the $R_b$ parameterisation used, which for forests can vary significantly". Please provide any suitable reference to this.

8) Lines 401-402: "For HONO and $NH_3$, respectively, 26% and 19% of calculated fluxes were positive, i.e. emissions." Is this indicative of anthropogenic and biogenic sources or either one of the two as predominant source (refer to comment # 5 and Lines 455-460: $BC_e$ were strongest for $NH_3$ ($r_s$= 0.60)….HONO……not as strongly correlated)? Any details on source characterization at ATTO site that might be helpful to explain it further? That might also explain more on: "why desorption would have been more important for HONO than for $NH_3$."(Line 697)

9) Lines 413-414: "From process-orientated modelling of aerosol $V_d$, it has been suggested that particle $V_d$ increases over increasingly rough surfaces." Provide suitable/recent references other than Gallagher et al. (2002) if possible.

10) Line 575 and Figure 15: Instead of 'Total suspended particulate' simply use 'inorganic particulates' to be more accurate? And keep that consistent throughout the manuscript.

**11)** Lines 730-742: As authors note that: this study is conducted around October 2017 only, it might be too conservative to assume 'annual dry deposition of total reactive nitrogen for the ATTO site to be 1.7kg-N ha$^{-1}$ a$^{-1}$ (i.e. same as for this study done in non-growing season for agriculture)'. Since, peak agricultural activity might be occurring between May-September. Clarifying more on time-period of Trebs et al. (2006) study that gives more than twice the total dry deposition value than this study would help.

**12)** Line 759: please correct typing error: "ABLE-2A of NO$^+_3$" to NO$_3^-$

---

## Author Comment (AC1) · 25 Oct 2020

**Authors' Response to Reviews of**

**Concentrations and biosphere-atmosphere fluxes of inorganic trace gases and associated ionic aerosol counterparts over the Amazon rainforest**

R. Ramsay et al.
*Atmospheric Chemistry and Physics,* `doi.org/10.5194/acp-2020-586`
* * *
**RC:** *Reviewers' Comment*,    AR: Authors' Response,    ☐ Manuscript Text

**1. Introduction**

We thank the reviewers for their time in reviewing our paper and providing comments. Below, we present our responses to each reviewer in turn, addressing each of their points individually. With some points, there is overlap between reviewers. Where this arises, we have included both comments, and provided a response.

**2. Reviewer #1**

**2.1. Revision 1**

**RC:** *Line 2-4: The wet season can also be affected by anthropogenic pollution, such as that from Manaus.*

AR: We thank the reviewer for noting this important point. We agree that it should be further stressed that the wet season is still impacted by anthropogenic sources of pollution, with clarification on the use of the term "near-pristine". However, we think that this would be better placed within the body of the text itself, rather than over complicate the abstract. We have therefore included the following explanation in **Section 2.1, line 111-112**: "The conditions are termed "near-pristine", as regional sources of anthropogenic pollution can still intrude at the site during this time period".

**2.2. Revision 2**

**RC:** *Line 354-360: What are the possible reasons for the large difference between GRAEGOR and ToF-ACSM measurements for nitrate and chloride?*

**RC:** *(from Reviewer # 2 ) Lines 348-360 and Figure 3: Authors mention there is a significant difference between GRAEGOR and ToF-ACSM measurements for both nitrate and chloride. Maybe consider adding any linear regression analysis as done for SO42- and NH4+ to reach that conclusion (to ensure it is not a scale issue, nitrate and chloride ToF-ACSM measurements being order of magnitudes lower than GRAE-GOR)? More discussion on how different sampling heights of GRAEGOR and ToF-ACSM measurements (60 m and 321 m respectively) matter, might help? Consider merging the argument made in Lines 427-429 to explain the GRAEGOR vs TOF-ACMS difference for : "...Vds with increasing particle size, the larger median Vds values for nitrate and chloride are consistent with the GRAEGOR vs ACSM comparison which suggests that these aerosol counterparts were present in the super-micron (>PM1) fraction."*

AR: As outlined in Section 4.3.2., we think that the difference between GRAEGOR and ToF-ACSM measurements

of nitrate and chloride aerosol is due to the fact that the ACSM measures only non-refractory aerosol in the sub-micron range, while the GRAEGOR measures total suspended particulate. The ACSM would therefore be insensitive to refractory seasalt and crustal material, which – from previous studies conducted at ATTO and from thermodynamic modelling using ISORROPIA-2 – we think is the source for the nitrate and chloride aerosol measured by the GRAEGOR. We have included cross-section references to Section 4.3.2 in Section 3.2. that clarifies the differences that exist, and why we think this exists independently of the sampling height.

**2.3. Revision 3**

**RC:** *Line 779-780: Trebs et al. (2004) shows an average $NH_3$ concentration of about 2 ppb in the dry season, which is about 4 times larger than the measurement of this study (about 0.5 ppb)*

**RC:** *(from Reviewer #2) Lines 730-742: As authors note that: this study is conducted around October 2017 only, it might be too conservative to assume 'annual dry deposition of total reactive nitrogenfor the ATTO site to be 1.7kg-N ha1a1(i.e. same as for this study done in non-growing season for agriculture)'. Since, peak agricultural activity might be occurring between May-September. Clarifying more on time-period of Trebs et al. (2006) study that gives more than twice the total dry deposition value than this study would help*

**AR:** We thank the reviewers for noticing both of these closely related issues. With regards to Reviewer 2's point, we have included the following in Section 4.5. : "The measurement period for the study by Trebs et al. (2006) occurred from 12 September 2002 to 14 November 2002. As noted by Trebs et al., measurements included in September occur during the tail end of the peak agricultural season in the Amazon Basin". With regards to the issue raised by Reviwer / 1, we have ammended Lines 786 onward: "Measurements in the dry season had similar mean and median concentrations of $HNO_3$ as this study, but higher concentrations of HCl, HONO and $NH_3$ (a mean concentration of 2 ppb $NH_3$ compared to 0.5 ppb $NH_3$ as measured by this study), and with $SO_2$ having the lowest concentration of the inorganic trace gases measured."

**2.4. Revision 4**

**RC:** *Many texts in the figures are quite small. You need to make sure that all texts are legible without zooming in.*

**AR:** We have revised the figures accordingly.

**2.5. Revision 5**

**RC:** *Figs. 2 and 3: You should also describe the black dashed lines and grey dashed lines in the figure captions.*

We thank the reviewer for noting this. We have amended the captions as follows – for figure 2, "...at 42 m (yellow,mass concentration; solid black line, molar mixing ratio) and 60 m (green, mass concentration; dashed grey line, molar mixing ratio)..."; for figure 3, "at 42 m (red, mass concentration; solid black line, molar mixing ratio) and 60 m (blue, mass concentration; dashed grey line, molar mixing ratio)"

**2.6. Revision 6**

**RC:** *Figs. 4 and 8: Please describe the meanings of each component of the boxplots.*

**AR:** We have added the following description to each boxplot caption – "The lower and upper edges of each box correspond to the first and third quartiles, while the whiskers extend to the largest and smallest values which

do not exceed 1.5 × the inter-quartile range from their respective hinge. Black dots outside the plots are values which exceed 1.5 × the inter-quartile range."

**2.7. Revision 7**

**RC:** *Fig. 6: This figure does not show pre-correction values as suggested by the caption.*

**RC:** *(from Reviewer #2) Figure 5: Please clarify in the caption that pre- and post- correction deposition velocity trends are denoted by colored and grey lines respectively or use same convention in Fig. 5 labels and caption.*

**AR:** We thank the reviewers for noting this error with Figure 6. Figure 6 only shows post corrected values for the deposition velocities of $NH_3$, $SO_2$ and HONO. We have now made this point clear in the caption.

**2.8. Revision 8**

**RC:** *Fig. 14: This figure is very similar to Fig. 1d except that an $SO_2$ line is added. I think it can be simply combined into Fig. 1. Also, I am wondering why the period before October 7 is not shown in Fig. 14.*

**RC:** *(from Reviewer #2) Please consider shifting some figures to supplement or split up some busy figures. For instance, if you want to keep the molar mixing ratios shown on secondary axes of Figs. 2 and 3, might be better to show them separately in supplement for clarity. Also increase text size of labels in figures wherever possible.*

**AR:** We thank the reviewers for noting where graphs could be removed. With regards to Figure 14 in the original manuscript, we believe that its inclusion is justified. This is a direct comparison between elemental black carbon and $SO_2$, and will be placed in the final manuscript close to the discussion section where this comparison is discussed. For the reader's aid, this visual comparison should be close to what is discussed. By amalgamating into Figure 1 or 2, we think that the impact is lost. In addition, Figure 1.d. currently has, on its secondary axis, the measured molar mixing ratio of CO, which is important to include as it is discussed in how periods are divided into polluted or non-polluted. Including $SO_2$ as part of Figure 1.d. would overburden the amount of data presented in this figure. For a similar reason (the inclusion of equivalent molar mixing ratios), it can not be added to Figure 2 either.

Reviewer 2 has made the suggestion that perhaps the molar mixing ratios could be moved to a supplemental. However, we believe that we should retain the use of these values, as expression concentrations as molar mixing ratios is the predominant method within the atmospheric science community, in comparison to mass concentrations. By including the molar mixing ratios in Figures 2 and 3, we hope to give the reader accustomed to molar mixing ratios a grasp of the magnitude of concentrations measured. However, we agree that there are too many figures, and as such, have moved Figures 12 and 13 of the original manuscript to a supplemental.

**2.9. Revision 9**

**RC:** *Fig. 15: I think "total suspended particulate" is not an appropriate term here because only a few inorganic components are included. Sometimes you used "water-soluble total suspended particulate" in the text. This is better but still not accurate since some organic components are also water-soluble.*

**RC:** *(from Reviewer #2) Line 575 and Figure 15: Instead of 'Total suspended particulate' simply use 'inorganic particulates' to be more accurate? And keep that consistent throughout the manuscript.*

AR:    We thank the reviewers for this suggestion. We agree with their assessment, and have revised Figure 15, and the overall manuscript, to refer instead to "inorganic particulate" instead of "total suspended particulate".

**3.   Reviewer #2**

**3.1.   Revision 1**

RC:    *As mentioned in the manuscript: "Based on the height of the tallest trees, the canopy height ($h_c$) is 37.5 m (Chor et al., 2017)." While, for this study two heights used for gradient measurements on the 80-m walk up tower were: $z_1$ = 42 m and $z_2$ = 60 m, both are above-canopy. Also Fig. 2 exhibits marginal difference in hourly concentrations measured at these two sampling heights above canopy. Can authors elaborate more on any limitations on doing measurements at a sampling height $< h_c$ (i.e. $< 37.5$ m)? Concentrations between above-canopy and below-canopy sampling heights would have shown more substantial gradient and possibly given better insights on canopy reductions and boundary layer chemistry of different species in a tropical rain forest? Is that something that can be focused in a future study (as discussed for HONO briefly in Lines 680-690)?*

AR:    Although it is correct that concentration differences between the two heights would be larger and easier to resolve if one were placed inside the canopy, this contravenes the principles of the aerodynamic gradient flux method used in this study. The flux-gradient relationship only holds above the canopy where the flux is constant with height. Inside the canopy the flux changes as vegetation layers and other elements start to act as sinks (or sources) of the compound that is being measured. There are alternative approaches, such as the inverse Lagrangian approach of Raupach (1989), which derive the flux from in-canopy concentration measurements, but these are conceptually more problematic, require a good understanding of the in-canopy turbulence and require measurements at more heights than the GRAEGOR can provide. As it is accepted standard methodology to make flux gradient measurements above the canopy, we do not see the need to clarify this further in the revised manuscript.

**3.2.   Revision 2**

RC:    *Please consider shifting some figures to supplement or split up some busy figures. For instance, if you want to keep the molar mixing ratios shown on secondary axes of Figs. 2 and 3, might be better to show them separately in supplement for clarity. Also increase text size of labels in figures wherever possible.*

RC:    *(from Reviewer #1) Fig. 14: This figure is very similar to Fig. 1d except that an $SO_2$ line is added. I think it can be simply combined into Fig. 1. Also, I am wondering why the period before October 7 is not shown in Fig. 14.*

AR:    We thank the reviewers for noting where graphs could be removed. With regards to Figure 14 in the original manuscript, we believe that its inclusion is justified. This is a direct comparison between elemental black carbon and $SO_2$, and will be placed in the final manuscript close to the discussion section where this comparison is discussed. For the reader's aid, this visual comparison should be close to what is discussed. By amalgamating into Figure 1 or 2, we think that the impact is lost. In addition, Figure 1.d. currently has, on its secondary axis, the measured molar mixing ratio of CO, which is important to include as it is discussed in how periods are divided into polluted or non-polluted. Including $SO_2$ as part of Figure 1.d. would overburden the amount of data presented in this figure. For a similar reason (the inclusion of equivalent molar mixing ratios), it can not be added to Figure 2 either.

Reviewer 2 has made the suggestion that perhaps the molar mixing ratios could be moved to a supplemental. However, we believe that we should retain the use of these values, as expressing concentrations as molar mixing ratios is the predominant method within the atmospheric science community, in comparison to mass concentrations. By including the molar mixing ratios in Figures 2 and 3, we hope to give the reader accustomed to molar mixing ratios a grasp of the magnitude of concentrations measured. However, we agree that there are too many figures, and as such, have moved Figures 12 and 13 of the original manuscript to a supplemental.

**3.3. Revision 3**

RC: *Lne 340: correct 'HNO)3' to HNO₃*

AR: We thank the reviewer for noticing this error. We have corrected it.

**3.4. Revision 4**

RC: *Lines 348-360 and Figure 3: Authors mention there is a significant difference between GRAEGOR and ToF-ACSM measurements for both nitrate and chloride. Maybe consider adding any linear regression analysis as done for SO42- and NH4+ to reach that conclusion (to ensure it is not a scale issue, nitrate and chloride ToF-ACSM measurements being order of magnitudes lower than GRAEGOR)? More discussion on how different sampling heights of GRAEGOR and ToF-ACSM measurements (60 m and 321 m respectively) matter, might help? Consider merging the argument made in Lines 427-429 to explain the GRAEGOR vs TOF-ACMS difference for : "...Vds with increasing particle size, the larger median Vds values for nitrate and chloride are consistent with the GRAEGOR vs ACSM comparison which suggests that these aerosol counterparts were present in the super-micron (>PM1) fraction."*

RC: *(from Reviewer # 1) Line 354-360: What are the possible reasons for the large difference between GRAE-GOR and ToF-ACSM measurements for nitrate and chloride?*

AR: As outlined in Section 4.3.2., we think that the difference between GRAEGOR and ToF-ACSM measurements of nitrate and chloride aerosol is due to the fact that the ACSM measures only non-refractory aerosol in the sub-micron range, while the GRAEGOR measures total suspended particulate. The ACSM would therefore be insensitive to refractory seasalt and crustal material, which – from previous studies conducted at ATTO and from thermodynamic modelling using ISORROPIA-2 – we think is the source for the nitrate and chloride aerosol measured by the GRAEGOR. We have included cross-section references to Section 4.3.2 in Section 3.2. that clarifies the differences that exist, and why we think this exists independently of the sampling height.

**3.5. Revision 5**

RC: *Lines 362-365: "Although the diel cycle of HONO exhibited a maximum during night and a minimum during the day, it remained above the detection limit even during daylight hours (Figure 4), which, given the high photolysis rate of HONO during daytime, implies the presence of a daytime source." Can this daytime source of HONO point to biogenic soil HONO emissions? How would they compare to anthropogenic sources in 'polluted' conditions?*

AR: As presented in Section 4.3.2 of the manuscript, we set out potential sources for HONO at the ATTO site. However, it is important, as the reviewer notes, to place the concentrations in context with urban sites. As such, we have added the following material to lines 369 - 372: "The measured mean concentration of HONO of this study is similar to measurements of HONO taken over rural and pristine areas (Spataro, 2014), but is below the 0.1 to 0.8 ppb values that are measured at some urban sites (Hendrick et al., 2014). We discuss the

potential sources for HONO at the ATTO field site in Section Section 4.4.2."

**3.6. Revision 6**

RC: *Figure 5: Please clarify in the caption that pre- and post- correction deposition velocity trends are denoted by colored and grey lines respectively or use same convention in Fig. 5 labels and caption.*

RC: *(from Reviewer #1) Fig. 6: This figure does not show pre-correction values as suggested by the caption.*

AR: We thank the reviewers for noting this error with Figure 6. Figure 6 only shows post corrected values for the deposition velocities of $NH_3$, $SO_2$ and HONO. We have now made this point clear in the caption.

**3.7. Revision 7**

RC: *Line 390: "......although the results would be sensitive to the $R_b$ parameterisation used, which for forests can vary significantly". Please provide any suitable reference to this*

AR: We have amended the manuscript to include the following reference which compares different parameterisations of $R_b$ with regards to forest ecosystems: Jensen, N. O. and Hummelshøj, P.: Derivation of canopy resistance for water vapour fluxes over a spruce forest, using a new technique for the viscous sublayer resistance, Agric. For. Meteorol., 73(3-4), 339–352, doi:10.1016/0168-1923(94)05083-I, 1995.

**3.8. Revision 8**

RC: *Lines 401-402: "For HONO and $NH_3$, respectively, 26% and 19% of calculated fluxes were positive, i.e. emissions." Is this indicative of anthropogenic and biogenic sources or either one of the two as predominant source (refer to comment 5 and Lines 455-460: BCe were strongest for $NH_3$ (rs= 0.60)....HONO......not as strongly correlated)? Any details on source characterization at ATTO site that might be helpful to explain it further? That might also explain more on: "why desorption would have been more important for HONO than for $NH_3$."(Line 697)*

AR: We belief that source characterisation is an important field to consider as a result of this study. We have attempted, in our response to Revision 5, to include some detail of source characterisation with regard to HONO. For $NH_3$, we analyse possible source contribution further in our upcoming paper (Ramsay et al 2020, https://doi.org/10.5194/bg-2020-219), currently in peer review.

**3.9. Revision 9**

RC: *Lines 413-414: "From process-orientated modelling of aerosol $V_d$, it has been suggested that particle $V_d$ increases over increasingly rough surfaces." Provide suitable references other than Gallagher et al. (2002) if possible.*

AR: We have included an additional reference to Gallagher et al., 2002; the Petroff et al., 2008b, which presents a new modelling approach to aerosol dry deposition. This builds on Gallagher's original linkage of surface roughness and deposition velocity.

**3.10. Revision 10**

RC: *Line 575 and Figure 15: Instead of 'Total suspended particulate' simply use 'inorganic particulates' to be more accurate? And keep that consistent throughout the manuscript.*

**RC:** *(from Reviewer #1) Fig. 15: I think "total suspended particulate" is not an appropriate term here because only a few inorganic components are included. Sometimes you used "water-soluble total suspended particulate" in the text. This is better but still not accurate since some organic components are also water-soluble.*

 **AR:** We thank the reviewers for this suggestion. We agree with their assessment, and have revised Figure 15, and the overall manuscript, to refer instead to "inorganic particulate" instead of "total suspended particulate".

**3.11. Revision 11**

**RC:** *Lines 730-742: As authors note that: this study is conducted around October 2017 only, it might be too conservative to assume 'annual dry deposition of total reactive nitrogen for the ATTO site to be 1.7kg-N ha1a1(i.e. same as for this study done in non-growing season for agriculture)'. Since, peak agricultural activity might be occurring between May-September. Clarifying more on time-period of Trebs et al. (2006) study that gives more than twice the total dry deposition value than this study would help.*

**RC:** *(from Reviewer #1) Line 779-780: Trebs et al. (2004) shows an average $NH_3$ concentration of about 2 ppb in the dry season, which is about 4 times larger than the measurement of this study (about 0.5 ppb)*

 **AR:** We thank the reviewers for noticing both of these closely related issues. With regards to Reviewer 2's point, we have included the following in Section 4.5. : "The measurement period for the study by Trebs et al. (2006) occurred from 12 September 2002 to 14 November 2002. As noted by Trebs et al., measurements included in September occur during the tail end of the peak agricultural season in the Amazon Basin". With regards to the issue raised by Reviwer / 1, we have ammended Lines 786 onward: "Measurements in the dry season had similar mean and median concentrations of $HNO_3$ as this study, but higher concentrations of HCl, HONO and $NH_3$ (a mean concentration of 2 ppb $NH_3$ compared to 0.5 ppb $NH_3$ as measured by this study), and with $SO_2$ having the lowest concentration of the inorganic trace gases measured."

**3.12. Revision 12**

**RC:** *R2, P12 - Line 759: please correct typing error: "ABLE-2A of $NO_3^+$" to $NO_3^-$*

 **AR:** We thank the reviewer for noting this error, which is now corrected in the current version of the manuscript.

[revised manuscript text omitted]
^{-3}$. We discuss the reasons for the discrepancy between ToF-ACSM and GRAEGOR measurements of $NO_3^-$ and $Cl^-$ further in Section 4.3.2.

The median ($0.06\,\mu g\,m^{-3}$, 0.03 ppb) and mean ($0.07\,\mu g\,m^{-3}$ 0.04 ppb) values for the inorganic trace gas nitrous acid (HONO) remained above the detection limit of the instrument ($30\,ng\,m^{-3}$) at both sampling heights. Although the diel cycle of HONO exhibited a maximum during night and a minimum during the day ($0.02\,\mu g\,m^{-3}$ at 14:00), it remained above the detection limit even during daylight hours (Figure 4), which, given the high photolysis rate of HONO during daytime, implies the presence of a daytime source. The measured mean concentration of HONO of this study is similar to measurements of HONO taken over rural and pristine areas (Spataro and Ianniello, 2014), but is below the 0.1 to 0.8 ppb values that are measured at some urban sites (Hendrick et al., 2014). We discuss the potential sources for HONO at the ATTO field site in Section 4.4.2.

[revised manuscript text omitted]
 measurement period for the study by (Trebs et al., 2006) occurred from 12 September
750  2002 to 14 November 2002. As noted by Trebs et al., measurements included in September occur during the tail end of the peak agricultural season in the Amazon Basin. The stronger influence of agricultural activities and closer proximity of biomass burning at the pasture site in the Trebs et al. study may explains the slightly higher total $\Sigma_{N_r}$.

**4.6 Comparisons of measured concentrations of trace gases and associated aerosols with previous studies**

Whilst this was a one-month study limited to the dry season, during which local, regional and global biomass burning con-
755  tributed to observed concentrations, it provides some insight into the atmospheric composition of an ecosystem for which there are few measurements overall. Placing these measurements in context with similar regional and local studies above tropical rainforest sites provides an impression of the spatial and temporal representativeness of this study.

For aerosols, measurements of $PM_{10}$ concentrations (both cations and anions) taken by high-volume air samplers between 2008 and 2016 over the Cuieiras ZF2 natural reserve approximately $130\,\mathrm{km}$ west of the ATTO site have recently become
760  available (Custodio et al., 2019), allowing a local comparison in measured aerosol concentrations between the GRAEGOR and filter sampling. For $Cl^-$ and $NO_3^-$, the average measurements taken by the GRAEGOR are between 2.5 and 4 times greater than the average from 10 samples collected by the high volume air samplers during the dry seasons in the period 2008 to 2016. Conversely, the average dry season $SO_4^{2-}$ concentrations recorded by the GRAEGOR is 0.3 times that recorded by the high volume samplers. $NH_4^+$ concentrations recorded by both measurement techniques are approximately equivalent.
765  Measurements of aerosol composition taken during the Amazon Boundary Layer Experiment (ABLE-2A) (Talbot et al., 1988) provide mean concentration values for the same species measured during this study. Talbot et al. measures a mean atmospheric concentration in the mixed layer for $NH_4^+$ as $12\,\mathrm{nmol^{-3}}$ or $0.22\,\mathrm{\mu g\,m^{-3}}$; and for $SO_4^{2-}$ as $5.2\,\mathrm{nmol^{-3}}$ or 0.5

$\mu g\,m^{-3}$. These values are higher than those measured in this study (mean concentration of $NH_4^+ = 0.16\,\mu g\,m^{-3}$, and $SO_4^{2-} = 0.25\,\mu g\,m^{-3}$). In comparison, the mean concentrations measured during ABLE-2A of $NO_3^-$ ($4.4\,nmol^{-3}$ or $0.22\,\mu g\,m^{-3}$) and

770    $Cl^-$ ($1.2\,nmol^{-3}$ or $0.04\,\mu g\,m^{-3}$) are lower than those measured during this study.

Discrepancies in the measurements of these aerosol species between wet-chemistry instruments and high-volume air-sampler systems have previously been noted by Trebs et al. (2008), who found a similar order of magnitude difference in $SO_4^{2-}$ measurements between a WRD-SJAC system and a high volume air sampler in tropical conditions. They also reported that high-volume air samplers measured lower concentrations of $Cl^-$ and $NO_3^-$ compared to wet chemistry instruments, although

775    this pattern was only observed during periods of low concentrations of $Cl^-$ and $NO_3^-$. Loss of $Cl^-$ and $NO_3^-$ from high volume filters has been reported frequently, and this issue in fact led to the development of the SJAC sampling system, which does not suffer from this artefact (Slanina et al., 2001). Trebs et al. (2008) attributed higher $SO_4^{2-}$ high-volume air-sampler concentrations to the decomposition of organosulfates on filters during storage, as well as to environmental conditions such as high relative humidity that may have introduced both positive and negative artefacts on the filter substrate.

780    The most comprehensive previous report of $NH_3$, $SO_2$ and $HNO_3$ concentrations over remote tropical rainforests is by Adon et al. (2010), who presented long-term measurements over Cameroonian rainforest using passive denuder tubes. For the dry season, Adon et al. reported a similar concentration of $SO_2$ and $HNO_3$, but reported a significantly higher concentration of $NH_3$ (a dry season average of $2.9\,\mu g\,m^{-3}$ compared to $0.28\,\mu g\,m^{-3}$ reported in this study). Adon et al. postulated that the $NH_3$ concentrations recorded over their rainforest site were driven by biomass burning, similar to the conclusion drawn in this

785    study. It is possible that the intensity, proliferation and proximity of biomass burning at the Cameroonian site may therefore be heightened in comparison to the ATTO site, resulting in greater measurements of $NH_3$ concentrations.

Trebs et al. (2004), using a wet annular rotating denuder with steam jet aerosol collector system – effectively a single-height GRAEGOR instrument – measured the same suite of inorganic trace gases and associated aerosols as this study, but at a pasture site located in the southern Amazon Basin. Measurements in the dry season had similar mean and median concentrations of

790    $HNO_3$ as this study, but higher concentrations of HCl, HONO and $NH_3$ (a mean concentration of 2 ppb NH3 compared to 0.5 ppb $NH_3$ as measured by this study), 
[revised manuscript text omitted]

George, C., Strekowski, R. S., Kleffmann, J., Stemmler, K., and Ammann, M.: Photoenhanced uptake of gaseous $NO_2$ on solid organic compounds: a photochemical source of HONO?, Faraday discussions, 130, 164–195,519–524, https://doi.org/10.1039/B417888M, 2005.

Gloor, M., Gatti, L., Brienen, R., Feldpausch, T. R., Phillips, O. L., Miller, J., Ometto, J. P., Rocha, H., Baker, T., de Jong, B., Houghton, R. A., Malhi, Y., Aragão, L. E. O. C., Guyot, J.-L., Zhao, K., Jackson, R., Peylin, P., Sitch, S., Poulter, B., Lomas, M., Zaehle, S., Hunt-
1005    ingford, C., Levy, P., and Lloyd, J.: The carbon balance of South America: a review of the status, decadal trends and main determinants, Biogeosciences, 9, 5407–5430, https://doi.org/10.5194/bg-9-5407-2012, 2012.

Graedel, T. E. and Keene, W. C.: Tropospheric budget of reactive chlorine, Global Biogeochemical Cycles, 9, 47–77, https://doi.org/10.1029/94GB03103, 1995.

He, Y., Zhou, X., Hou, J., Gao, H., and Bertman, S. B.: Importance of dew in controlling the air-surface exchange of HONO in rural forested environments, Geophysical Research Letters, 33, https://doi.org/10.1029/2005GL024348, 2006.

Hendrick, F., Clémer, K., Wang, P., De Mazière, M., Fayt, C., Gielen, C., Hermans, C., Ma, J., Pinardi, G., Stavrakou, T., Vlemmix, T., and Van Roozendael, M.: Four years of ground-based MAX-DOAS observations of HONO and NO2 in the Beijing area, Atmospheric Chemistry and Physics, 14, 765–781, https://doi.org/10.5194/acp-14-765-2014, 2014.

Holanda, B. A., Pöhlker, M. L., Walter, D., Saturno, J., Sörgel, M., Ditas, J., Ditas, F., Schulz, C., Franco, M. A., Wang, Q., Donth, T., Artaxo, P., Barbosa, H. M. J., Borrmann, S., Braga, R., Brito, J., Cheng, Y., Dollner, M., Kaiser, J. W., Klimach, T., Knote, C., Krüger, O. O., Fütterer, D., Lavrič, J. V., Ma, N., Machado, L. A. T., Ming, J., Morais, F. G., Paulsen, H., Sauer, D., Schlager, H., Schneider, J., Su, H., Weinzierl, B., Walser, A., Wendisch, M., Ziereis, H., Zöger, M., Pöschl, U., Andreae, M. O., and Pöhlker, C.: Influx of African biomass burning aerosol during the Amazonian dry season through layered transatlantic transport of black carbon-rich smoke, Atmospheric Chemistry and Physics, 20, 4757–4785, https://doi.org/10.5194/acp-20-4757-2020, 2020.

Jardine, K., Yañez-Serrano, A. M., Williams, J., Kunert, N., Jardine, A., Taylor, T., Abrell, L., Artaxo, P., Guenther, A., Hewitt, C. N., House, E., Florentino, A. P., Manzi, A., Higuchi, N., Kesselmeier, J., Behrendt, T., Veres, P. R., Derstroff, B., Fuentes, J. D., Martin, S. T., and Andreae, M. O.: Dimethyl sulfide in the Amazon rain forest, Global Biogeochemical Cycles, 29, 19–32, https://doi.org/10.1002/2014GB004969, 2015.

Jensen, N. and Hummelshøj, P.: Derivation of canopy resistance for water vapour fluxes over a spruce forest, using a new technique for the viscous sublayer resistance, 
[revised manuscript text omitted]

Spataro, F. and Ianniello, A.: Sources of atmospheric nitrous acid: State of the science, current research needs, and future prospects, Journal of the Air & Waste Management Association, 64, 1232–1250, https://doi.org/10.1080/10962247.2014.952846, 2014.

Spindler, G., Hesper, J., Brüggemann, E., Dubois, R., Müller, T., and Herrmann, H.: Wet annular denuder measurements of nitrous acid: laboratory study of the artefact reaction of $NO_2$ with S(IV) in aqueous solution and comparison with field measurements, Atmospheric Environment, 37, 2643–2662, https://doi.org/https://doi.org/10.1016/S1352-2310(03)00209-7, 2003.

Stein, A. F., Draxler, R. R., Rolph, G. D., Stunder, B. J. B., Cohen, M. D., and Ngan, F.: NOAA's HYSPLIT Atmospheric Transport and Dispersion Modeling System, Bulletin of the American Meteorological Society, 96, 2059–2077, https://doi.org/10.1175/BAMS-D-14-00110.1, 2015.

Stemmler, K., Ndour, M., Elshorbany, Y., Kleffmann, J., D'Anna, B., George, C., Bohn, B., and Ammann, M.: Light induced conversion of nitrogen dioxide into nitrous acid on submicron humic acid aerosol, Atmos. Chem. Phys., 7, 4237–4248, https://doi.org/10.5194/acp-7-4237-2007, 2007.

Su, H., Cheng, Y., Oswald, R., Behrendt, T., Trebs, I., Meixner, F. X., Andreae, M. O., Cheng, P., Zhang, Y., and Pöschl, U.: Soil Nitrite as a Source of Atmospheric HONO and OH Radicals, Science, 333, 1616 LP – 1618, https://doi.org/10.1126/science.1207687, 2011.

Sullivan, R. C., Guazzotti, S. A., Sodeman, D. A., Tang, Y., Carmichael, G. R., and Prather, K. A.: 
[revised manuscript text omitted]

[Figure]

**Figure 3.** Time series of hourly concentrations (primary left axis, mass concentrations; secondary right axis, molar mixing ratios) of water-soluble aerosol species measured by the GRAEGOR at 42 m (red, mass concentration; solid black line, molar mixing ratio) and 60 m (blue, mass concentration; dashed grey line, molar mixing ratio) on the 80-m tower, and ToF-ACSM at 321 m (black) at the Amazon Tall Tower, at the Amazon Tall Tower Observatory site.

[Figure]

**Figure 4.** Median hourly diel concentrations for the inorganic trace gases $NH_3$, $SO_2$, HONO, $HNO_3$ and HCl in blue, and their paired associated aerosol counterparts $NH_4^+$, $SO_4^{2-}$, $NO_3^-$ and $Cl^-$ in red at the 60 m sampling height measured during the campaign. The lower and upper edges of each box correspond to the first and third quartiles, while the whiskers extend to the largest and smallest values which do not exceed 1.5 $\times$ the inter-quartile range from their respective hinge. Black dots outside the plots are values which exceed 1.5 $\times$ the inter-quartile range.

[Figure]

**Figure 5.** Inorganic trace gas deposition velocities ($V_d$) pre- and post- correction with $\gamma_F$ (Chor et al., 2017) and calculated theoretical maximum deposition velocities ($V_{max}$) for HCl and HNO$_3$.

[Figure]

**Figure 6.** Inorganic trace gas deposition velocities ($V_d$) (post- correction) with $\gamma_F$ (Chor et al., 2017) and calculated theoretical maximum deposition velocities ($V_{max}$) for $NH_3$, $SO_2$ and HONO.

[Figure]

**Figure 7.** Time series of filtered fluxes for the inorganic trace gas species measured during the campaign.

[Figure]

**Figure 8.** Calculated median diel fluxes of inorganic trace gas species measured during the campaign. From top left (clockwise) – HCl, $HNO_3$, $SO_2$, HONO and $NH_3$. The lower and upper edges of each box correspond to the first and third quartiles, while the whiskers extend to the largest and smallest values which do not exceed $1.5 \times$ the inter-quartile range from their respective hinge. Black dots outside the plots are values which exceed $1.5 \times$ the inter-quartile range.

[Figure]

**Figure 9.** Time series of filtered fluxes for the aerosol counterpart species measured during the campaign.

[Figure]

**Figure 10.** Fractional contribution to total measured inorganic acidity from $SO_2$, $HNO_3$, HONO and HCl as measured by the GRAEGOR at 60 m (hourly resolution). The concentration of inorganic particulate $Cl^-$ is included as an indicator of periods where sea salt or chloride containing particulate was present at the ATTO site.

[Figure]

**Figure 11.** Air-mass back-trajectories arriving at the 80-m walk up tower on each day every three hours from 00:00 local time over the period from 6 October 2017 to 3 November 2017, grouped by week, and coupled with associated frequency trajectory plots. The duration of each trajectory is 10 days, marks indicate 12-hour intervals. Modelled using NOAA HYSPLIT 4 using GDAS1 meteorology.

[Figure]

**Figure 12.** Time series of hourly $SO_2$ and $BC_e$ concentrations, highlighting the close correlation between $SO_2$ and $BC_e$ measurements throughout the campaign.

[Figure]

**Figure 13.** Summed mass and speciation of inorganic particulate recorded by the GRAEGOR at 80-m throughout period of campaign.

[Figure]

**Figure A1.** Air-mass back-trajectories arriving at the 80-m walk up tower on each day every three hours from 00:00 local time over the period from 6 October 2017 to 31 October 2017, grouped by week, and further subdivided by day, for the regional area surrounding the ATTO site. Fire count data is included as an overlay to each weekly plot, with fire count coloured according to the date on which the fire was recorded by satellite imagery.

[Figure]

**Figure A2.** Concentration weighted trajectory analysis for (from left) $BC_e$, $SO_4^{2-}$ and $NH_4^+$, with fire data overlaid. Fire data is coloured (scale, from light grey to black) by fire intensity, a measure of the fire radiative power of the individual fire.